# Deletion of Double Copies of the US1 Gene Reduces the Infectivity of Recombinant Duck Plague Virus *In Vitro* and *In Vivo*

Ying Wu,[a,b,c] Silun Tan,[a,b,c] Qing He,[a,b,c] Mingshu Wang,[a,b,c] Shun Chen,[a,b,c] Renyong Jia,[a,b,c] Qiao Yang,[a,b,c] Dekang Zhu,[b,c] Mafeng Liu,[a,b,c] Xinxin Zhao,[a,b,c] Shaqiu Zhang,[a,b,c] Juan Huang,[a,b,c] Xumin Ou,[a,b,c] Sai Mao,[a,b,c] Qun Gao,[a,b,c] Di Sun,[a,b,c] Bin Tian,[a,b,c] Anchun Cheng[a,b,c]

[a]Institute of Preventive Veterinary Medicine, Sichuan Agricultural University, Wenjiang, China

[b]Avian Disease Research Center, College of Veterinary Medicine of Sichuan Agricultural University, Wenjiang, China

[c]Key Laboratory of Animal Disease and Human Health of Sichuan Province, Sichuan Agricultural University, Wenjiang, China

Ying Wu, Silun Tan, and Qing He contributed equally to this work. Author order depended on the degree of responsibility for the article.

**ABSTRACT** Duck plague caused by duck plague virus (DPV) is one of the main diseases that seriously endangers the production of waterfowl. DPV possesses a large genome consisting of 78 open reading frames (ORFs), and understanding the function and mechanism of each encoded protein in viral replication and pathogenesis is the key to controlling duck plague outbreaks. US1 is one of the two genes located in the repeat regions of the DPV genome, but the function of its encoded protein in DPV replication and pathogenesis remains unclear. Previous studies found that the US1 gene or its homologs exist in almost all alphaherpesviruses, but the loci, functions, and pathogenesis of their encoded proteins vary among different viruses. Here, we aimed to define the roles of US1 genes in DPV infection and pathogenesis by generating a double US1 gene deletion mutant and its revertant without any mini-F cassette retention. *In vitro* and *in vivo* studies found that deletion of both copies of the US1 gene significantly impaired the replication, gene expression, and virulence of DPV, which could represent a potential candidate vaccine strain for the prevention of duck plague.

**IMPORTANCE** Duck plague virus contains nearly 80 genes, but the functions and mechanisms of most of the genes have not yet been elucidated, including those of the newly identified immediate early gene US1. Here, we found that US1 deletion reduces viral gene expression, replication, and virus production both *in vitro* and *in vivo*. This insight defines a fundamental role of the US1 gene in DPV infection and indicates its involvement in DPV transcription. These results provide clues for the study of the pathogenesis of the US1 gene and the development of attenuated vaccines targeting this gene.

**KEYWORDS** duck plague virus, US1, ICP22, replication, transcription, pathogenesis, *in vitro*, *in vivo*

Address correspondence to Anchun Cheng, chenganchun@vip.163.com.

The authors declare no conflict of interest.

Duck plague virus (DPV), alternatively named duck enteritis virus (DEV) or anatid herpesvirus 1 (AHV-1), is the causative agent of an acute septic infectious disease that can infect ducks, geese, swans, and other Anseriformes. Due to its fast transmission, wide spread, and high morbidity and mortality characteristics, it has now become one of the main diseases seriously endangering the production of waterfowl (1). DPV belongs to the *Herpesviridae*, *Alphaherpesvirus* subfamily, and *Mardivirus* genus. The virions of DPV are round with a diameter of approximately 160 to 180 nm (2) and consist of a linear double-stranded DNA genome, capsid, tegument, and envelope (3). Sequencing and comparative analysis of the DPV strains demonstrated that the Chinese virulent strain (CHv) of DPV has the longest genome to date, which contains 162,175 bases and contains approximately 78

genes (4, 5). Understanding the function and mechanism of each protein encoded by DPV in viral replication and pathogenesis is the key to effectively controlling duck plague.

Previous studies have demonstrated that herpesvirus genes are expressed in a temporal cascade during lytic infection, starting with immediate early (IE) genes, followed by early (E) and late (L) genes (6). IE genes are regarded as necessary for transactivating the transcription of other kinetic classes of genes. According to reports, US1 is an IE gene in herpes simplex virus 1 (HSV-1), herpes simplex virus 2 (HSV-2), varicella-zoster virus (VZV), and DPV (7–9) but is not an IE gene in pseudorabies virus (PRV) (10) and is both an IE gene and an L gene in bovine herpesvirus 1 (BoHV-1) and equid herpesvirus 1 (EHV-1) (11–13). Recent studies have found that the US1 gene or its homologs exist in many herpesviruses, normally encoding a multifunctional protein named infected-cell protein 22 (ICP22) or IE63 (14, 15). Studies of HSV-1 ICP22 found that it has a transcriptional regulatory function that can directly interact with cyclin-dependent kinase 9 (CDK9) and inhibit the enzyme activity of CDK9, thereby inhibiting polymerase II (Pol II) Ser-2 phosphorylation and keeping Pol II in a state of transcriptional pause, ultimately inhibiting the transcription of cellular and viral genes (16–18). In addition, it can recruit FACT (facilitates chromatin transcription) complexes to the viral genome, thereby stimulating Pol II to cross the nucleosome barrier and achieve efficient transcriptional elongation (19–21). It is also an important nonessential protein in the nuclear budding of HSV-1 virions by interacting with UL31 and/or UL34 (22, 23). The latest research also found that HSV-1 ICP22 is involved in virus immune evasion: HSV-1 ICP22 downregulates the expression of the costimulatory molecule CD80 in dendritic cells (DCs), thereby weakening the host immune response (24, 25); HSV-1 ICP22 also has the ability to relocate cellular heat shock protein 70 (Hsc70) to the virus-induced chaperone-enriched (VICE) domain in the nucleus, which prevents Hsc70 from exerting antiviral activity in other parts of the cell and weakens the host immune response (26). In herpes simplex virus 2 (HSV-2), the main function of ICP22 is related to the antihost antiviral response, and HSV-2 ICP22 suppresses the host antiviral response by inhibiting two phases of the type I interferon (IFN) response. HSV-2 ICP22 can interact with IRF-3, a key transcription factor in the production pathway of type I interferon (IFN). It inhibits the binding of IRF-3 and IFN-$\beta$, thereby reducing the expression of IFN-$\beta$ (9, 27–30); HSV-2 ICP22 can also act as an E3 ubiquitin ligase to induce the ubiquitination and degradation of signal transducer and activator of transcription 1 (STAT1), signal transducer and activator of transcription 2 (STAT2), and IFN regulatory 9 (IRF9), block type I IFN signaling, and then inhibit the production of IFN-stimulated genes (ISGs) (30). IE63 (the ICP22 homolog) encoded by VZV open reading frame 63/70 (ORF63/70) has also been found to have some functions, such as transcriptional regulation (31–33), establishment of latency (34, 35), and inhibition of host antiviral responses (36).

The DPV genome includes unique long (UL) and unique short (US) regions flanked by terminal and inverted repeat long (TRL/IRL) and short (TRS/IRS) regions, respectively (37, 38). To date, the US1 gene is known to be one of the two genes located in the repeat regions of the DPV genome, which results in two coexisting copies of the US1 gene; the other duplicated gene is ICP4. However, the US1 genes of HSV-1, HSV-2, and gallid herpesvirus 2 (GaHV-2) are located in the US region only as single-copy genes. To define the roles of US1 genes in DPV replication, gene expression, and pathogenesis, we generated a double US1 gene deletion mutant (2ΔUS1) and its revertant (2ΔUS1R) without any mini-F cassette retention based on previously established BAC-2ΔUS1 for further characterization *in vitro* and *in vivo* and evaluated its pathogenesis in infected ducklings. We found that 2ΔUS1 significantly impaired the replication, gene expression, and virulence of DPV *in vitro* and *in vivo*, which suggests that it could be a promising candidate vaccine strain that could help control duck plague outbreaks.

## RESULTS

**Generation of 2ΔUS1 and its revertant virus.** In earlier studies, we deleted both copies of the US1 gene one by one based on the bacterial artificial chromosome (BAC)

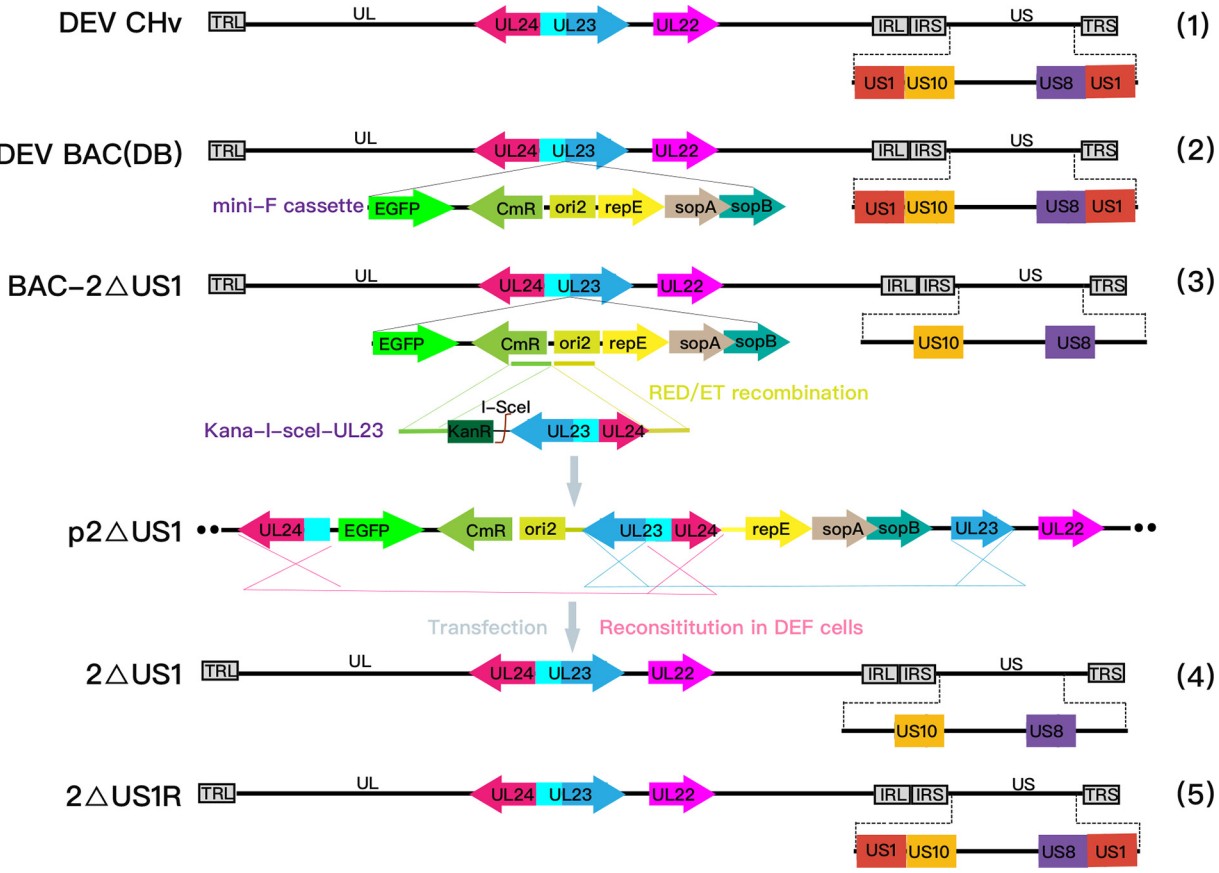

**FIG 1** Overview of the DEV wild-type strain and its derived recombinant virus genome structures. Line 1, wild-type DEV (CHv strain) genome; line 2, genome of the DEV bacterial artificial chromosome (DB); line 3, recombinant virus with deletion of double copies of the US1 gene based on DEV BAC (BAC-2ΔUS1); lines 4 and 5, US1 gene deletion mutant (2ΔUS1) and its revertant virus (2ΔUS1R) without mini-F elements. EGFP, enhanced green fluorescent protein.

rescue platform of the DPV CHv strain and found that the US1 gene encodes a nonessential IE protein located in the nucleus that plays a role in viral replication *in vitro* (7, 39). However, generation of the derived BAC-2ΔUS1 resulted in the retention of a minimal fertility factor replicon (mini-F) cassette (the backbone of the BAC vector) in the thymidine kinase (TK) (encoded by the UL23 gene) region of the DPV genome, which might mask the phenotype, either reducing or increasing the effect of US1 deletion, since the DPV genome could not be detected in ducks after parental DPV-BAC (DB) inoculation (unpublished data). We attributed this result to damage to the TK gene or the residue of the mini-F cassette in the DPV genome. To characterize US1 function *in vivo*, we inserted a 2.7-kb fragment containing Kana-I-SceI and TK (UL23) sequences between the bacterial origin of replication (ori2) and CmR to create an inverse duplication of DPV TK sequences adjoining the mini-F replicon for removing the mini-F cassette from the BAC-2ΔUS1 genome (Fig. 1). After two-step recombination in *Escherichia coli*, we obtained the 2ΔUS1 recombinant plasmid under steady chloramphenicol selection. Its revertant virus 2ΔUS1R was subsequently obtained to exclude nontarget mutations caused by genetic engineering manipulation. PCR and sequencing identification indicated that there was no difference in genetic composition between 2ΔUS1R and its wild-type strain CHv (data not shown).

After plasmid extraction, the plasmids 2ΔUS1 and 2ΔUS1R were transfected into duck embryo fibroblast (DEF) cells to reconstitute the recombinant virus. Due to an inverted duplication of DPV genomic sequences within the mini-F cassette, the markerless excision of vector sequences upon virus reconstitution triggered by the recombinases in eukaryotic cells occurs during intra- and intermolecular recombination events (Fig. 1). As shown in Fig. 2B,

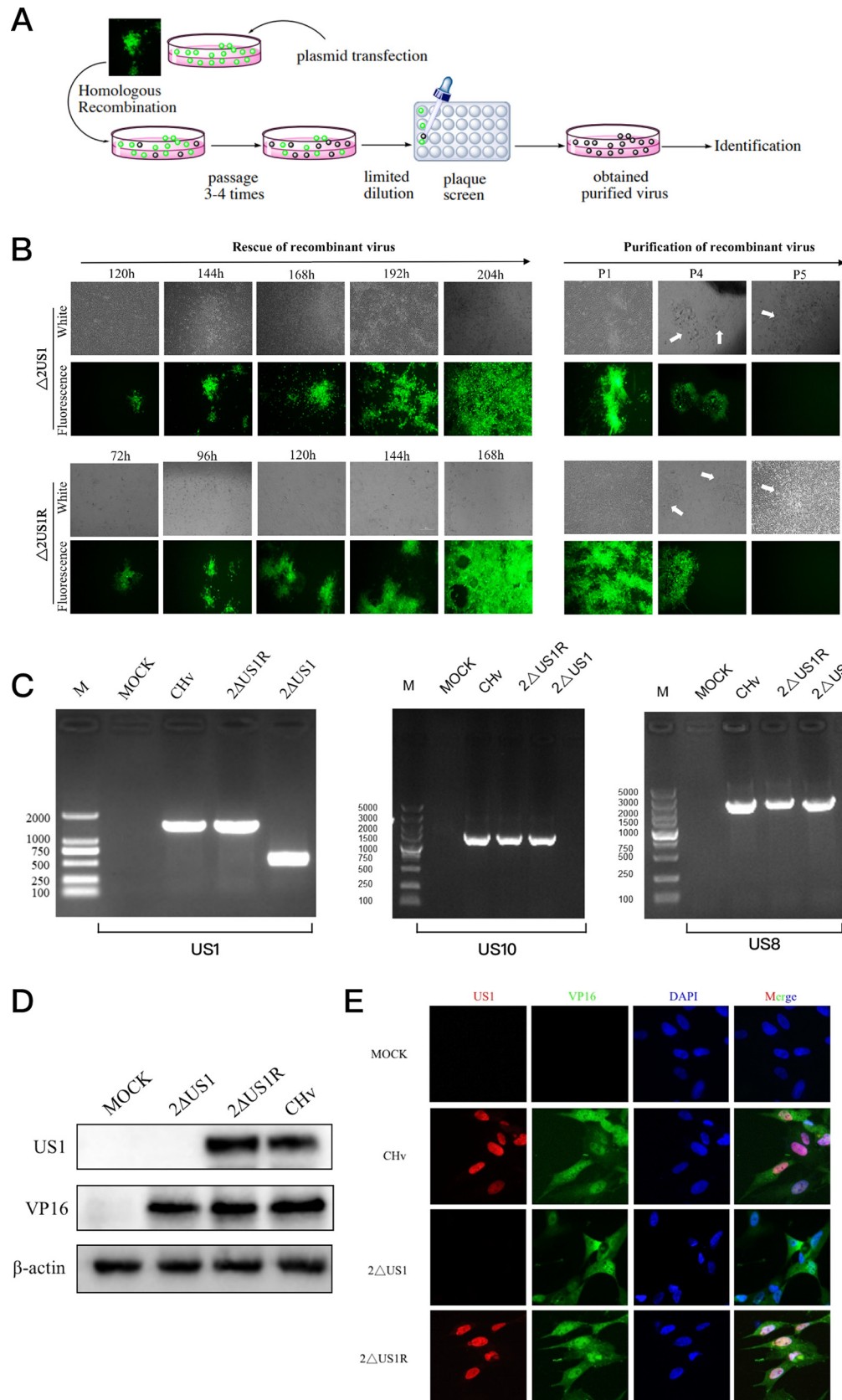

**FIG 2** Generation of 2ΔUS1 and its revertant 2ΔUS1R. (A) Schematic diagram of 2ΔUS1 and 2ΔUS1R construction and purification. (B) Rescue and purification of 2ΔUS1 and 2ΔUS1R on duck embryo fibroblast (DEF) cells. (C to E) PCR (C), Western blotting (D), and immunofluorescence assay (E) of 2ΔUS1 and 2ΔUS1R.

shortly after transfection, fluorescent spots began to appear and then spread throughout the field. Over time, the inverted duplication of DPV genomic sequences inside the mini-F element underwent homologous recombination triggered by the recombinases in DEF cells for the markerless excision of vector sequences from the DPV genome, resulting in the gradual disappearance of fluorescence. After plaque purification, the recombinant virus without fluorescence was obtained at passage 5 (P5). PCR amplification was first performed to confirm the US1 deletion and restoration. Mock-infected cells were used as a negative control. As shown in Fig. 2C, we found that there was a loss of nearly 1,000 bp in the 2ΔUS1 strain, meaning that the US1 genes were totally removed. The subsequent sequencing results confirmed the loss of the US1 gene in the DPV genome. Meanwhile, the integrity of the neighboring genes US10 and US8 was also tested by PCR and sequencing to confirm that the phenotype variation resulted from US1 deletion instead of US10 or US8 mutation. As a result, no mutations of the US10 and US8 genes have been found by PCR and sequencing. However, the expression of US10 and US8 genes in the 2ΔUS1 strain decreased significantly at the early stage of infection compared with that of the wild-type strain (data not shown). To rule out the effect of US8 and US10 on viral gene expression in 2ΔUS1, we transfected the US8/US10/US8+US10 plasmids before 2ΔUS1 infection. As a result, the overexpression of US10/US8/US10+US8 genes could not rescue the reduction of viral gene expression caused by US1 (data not shown). Taken together, we believe the reduction of viral genes can be attributed to the transcriptional regulation of the US1 gene rather than the reduction of US10 and US8. Further confirmation of US1 deletion and reversion at the protein level was carried out by Western blotting (WB) and immunofluorescence assays (IFAs). DEFs infected with CHv, 2ΔUS1, or 2ΔUS1R or mock infected were used for the detection of US1 protein. The housekeeping gene beta-actin was used as an internal control for each sample in WB, while the DPV VP16 gene was used to indicate virus infection in WB and IFA. As expected, no bands were detected in either mock- or 2ΔUS1-infected cells, while the US1 protein was detected in 2ΔUS1R- and CHv-infected cells (Fig. 2D). The result of IFA was similar to that of WB, and US1-specific fluorescence was detected in CHv- and 2ΔUS1R-infected cells but not in 2ΔUS1- or mock-infected cells (Fig. 2E). These results suggest that the 2ΔUS1 and 2ΔUS1R recombinant viruses were successfully constructed and could be used for subsequent experiments.

**In vitro characterization of 2ΔUS1 and 2ΔUS1R.** The replication and plaque morphology of each recombinant virus *in vitro* were characterized as described in Materials and Methods. Viral titers were calculated at the indicated time points after recombinant virus infection at a high (2) or low (0.01) multiplicity of infection (MOI), and the proliferation of the 2ΔUS1 and 2ΔUS1R strains was compared to that of the wild-type strain (CHv). As expected, the one-step and multistep growth curves of CHv and 2ΔUS1R were similar and displayed no significance at each time point. In contrast, the titer of 2ΔUS1 at later time points was apparently lower than that of CHv, which indicated that loss of the US1 gene impairs DPV replication *in vitro* (Fig. 3A). To test the effect of US1 gene deletion on viral DNA (vDNA) replication, the genomes of each recombinant virus were extracted and quantified using Taq-Man real-time quantitative PCR (RT-qPCR) by probing the UL30 gene. As shown in Fig. 3B, compared to CHv, loss of US1 apparently decreased the genomic DNA replication ability of DPV, but infection with a low dose of 2ΔUS1 (0.01 MOI) led to a greater reduction in the copy number of the DPV genome. This can be attributed to the fact that low-MOI infection with 2ΔUS1 will result in multiple rounds of infection, resulting in the amplification of the US1 gene's inhibitory effect on DPV genome replication. As expected, no significant difference in vDNA was observed between CHv and 2ΔUS1R. Plaque formation analysis results revealed that 2ΔUS1 caused apparently smaller (40%) and fewer (67%) plaques, while the spread of 2ΔUS1R in DEF cultures was comparable to that seen with the wild type (CHv) (Fig. 3D). These results suggest that US1 deletion significantly impairs viral yields and vDNA replication *in vitro*. Next, the impact of US1 deletion on DPV gene expression at the mRNA level was evaluated by real-time quantitative PCR (RT-qPCR).

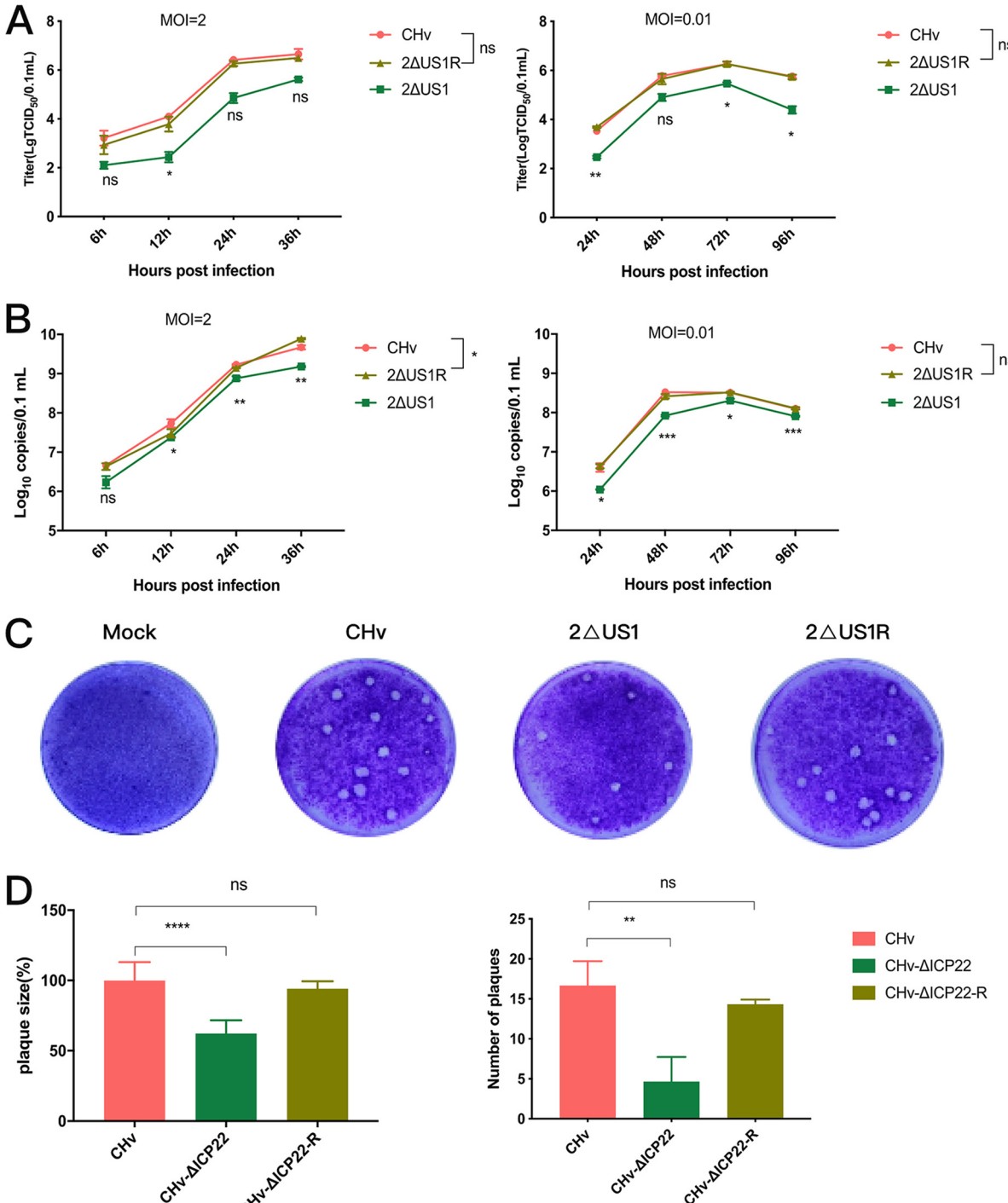

**FIG 3** Loss of US1 resulted in decreases in viral yields and vDNA replication *in vitro*. (A) Viral yields in duck embryo fibroblast (DEF) cells at an MOI of 2 or 0.01. (B) vDNA replication in DEF cells at an MOI of 2 or 0.01 measured by genomic DNA qPCR probing for the viral polymerase gene UL30. (C) Plaque formation of the indicated recombinant viruses in DEFs at an MOI of 0.0001. (D) Quantification analysis of data in panel C. The results represent mean values with error bars showing the standard error of the mean. Comparisons between CHv and 2ΔUS1/2ΔUS1R were analyzed using two-way ANOVA (A) and one-way ANOVA (B) and were considered significant as follows: *, $P < 0.05$; **, $P < 0.01$; ***, $P < 0.001$; ****, $P < 0.0001$; ns, not significant. Plots are representative of data from at least three independent experiments.

The results showed that the mRNA expression levels of the tested DPV genes in DEFs were significantly lower at the indicated time points after US1 deletion than after CHv deletion, but there was no difference in viral gene expression between 2ΔUS1R and CHv (Fig. 4), indicating that the US1 gene can promote the transcription of representative

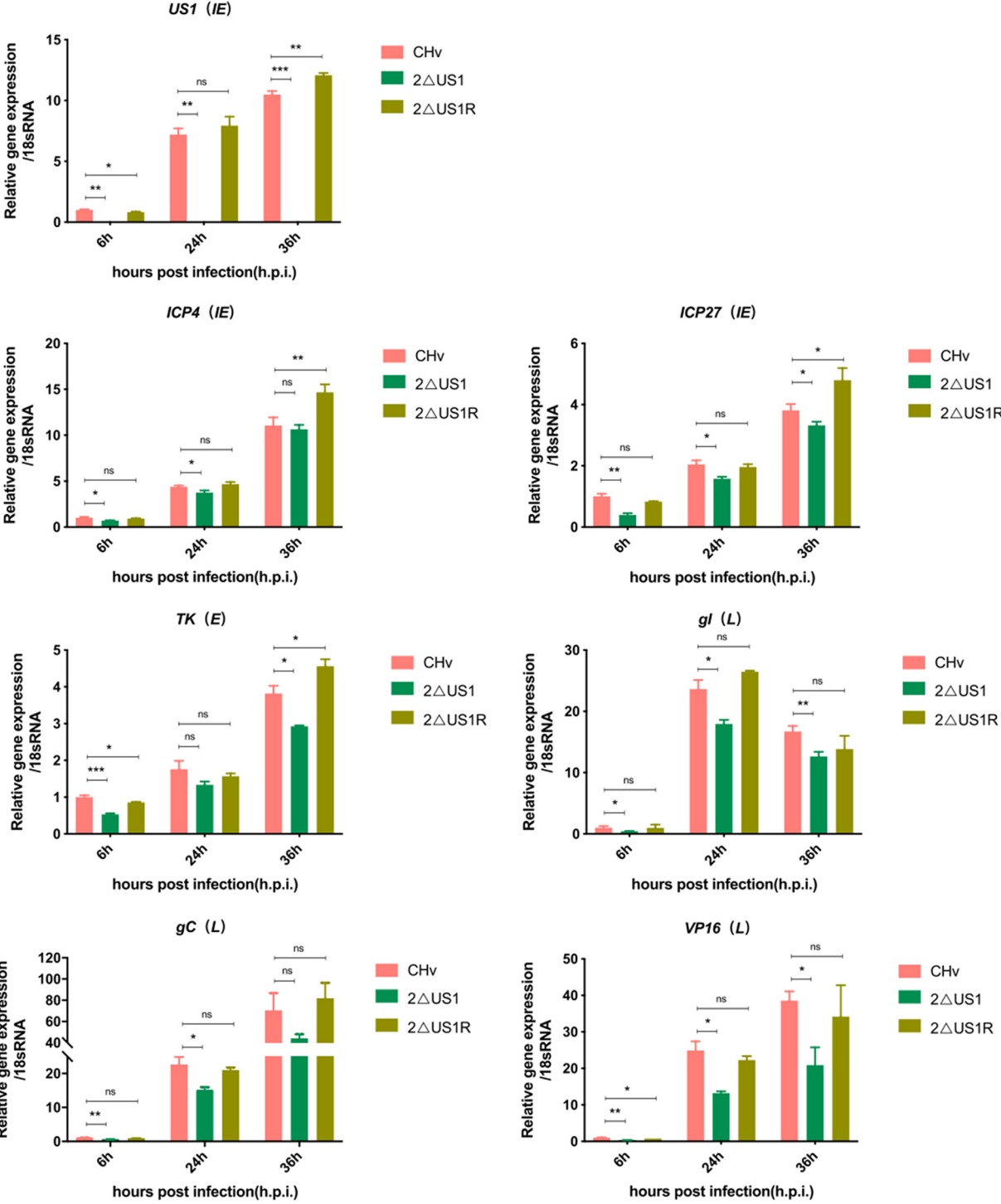

**FIG 4** *In vitro* effect of US1 deficiency on DPV gene expression. DPV viral gene transcription in DEF cells was measured by RT-qPCR normalized to duck 18S RNA. The results represent mean values with error bars showing the standard error of the mean. Comparisons between CHv and 2ΔUS1/2ΔUS1R were analyzed using two-way ANOVA and were considered significant as follows: *, $P < 0.05$; **, $P < 0.01$; ***, $P < 0.001$; ****, $P < 0.0001$; ns, not significant. Plots are representative of data from at least three independent experiments.

DPV IE (ICP4 and ICP27), E (TK), and L (gI, gC, and VP16) genes during viral infection (Fig. 4).

**2ΔUS1 of DPV was attenuated in ducks.** An *in vivo* experiment was subsequently performed to assess the effect of US1 on viral pathogenesis, and the flow chart of the animal experiment is shown in Fig. 5A. In group 1, 40 ducks inoculated with either

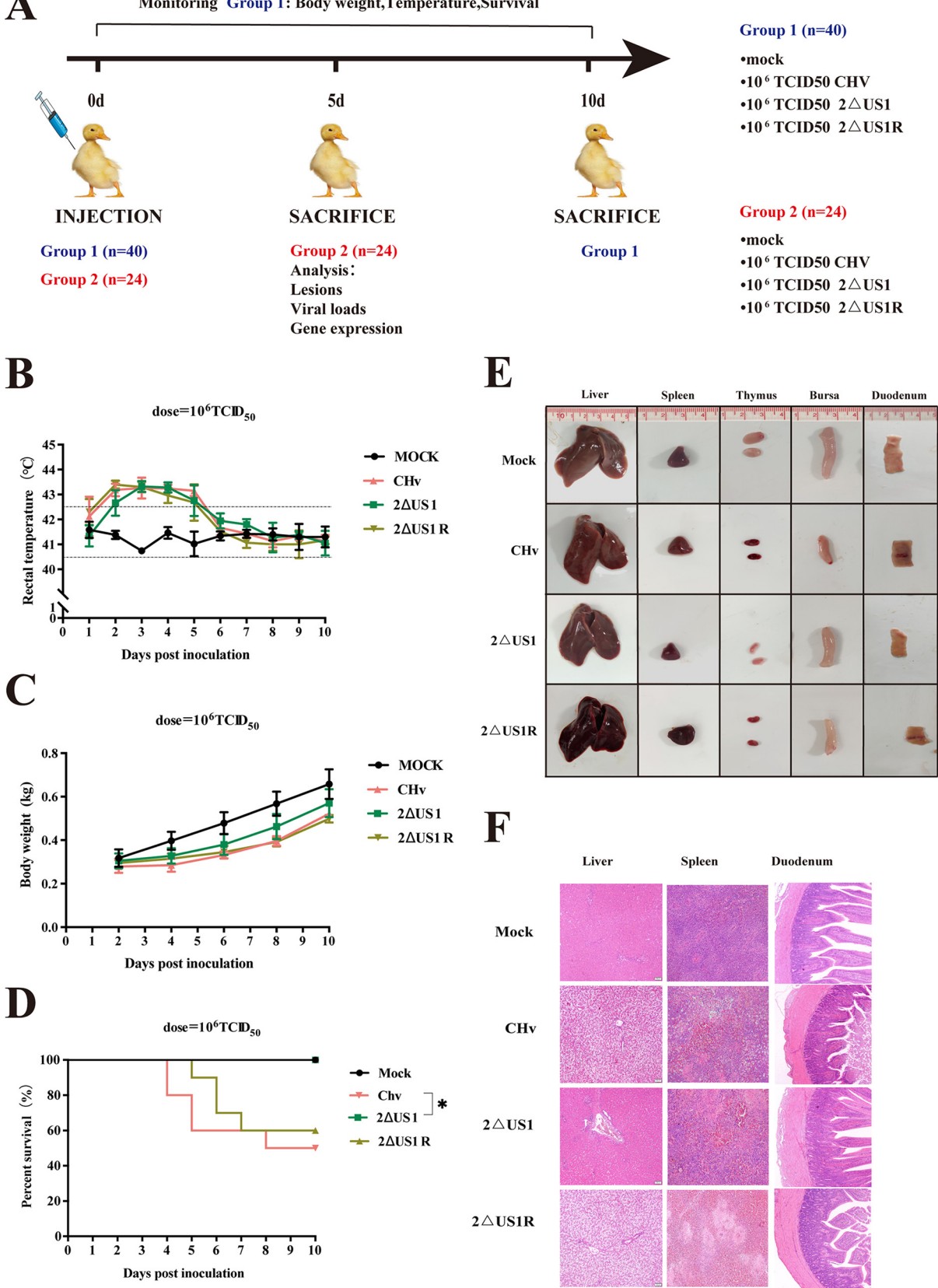

**FIG 5** The US1 gene contributes to the pathogenesis of DPV *in vivo*. (A) Flow chart of experiments performed to study the effect of 2ΔUS1 on DPV *in vivo*. Fourteen-day-old ducks were infected intranasally with CHv, 2ΔUS1, and 2ΔUS1R strains ($10^6$ TCID$_{50}$). Mock-infected ducks were used as a

wild-type CHv, 2ΔUS1, 2ΔUS1R, or minimal essential medium (MEM) were monitored for 10 days for survival and mortality. As shown in Fig. 5B to D, the ducklings inoculated with MEM for a blank control had no mortality and maintained normal body temperature and weight gain throughout the observation period, while the ducklings injected with CHv, 2ΔUS1, or 2ΔUS1R showed increased body temperature and decreased weight gain compared to the control. The survival curve of ducklings showed that inoculation with wild-type CHv and revertant virus 2ΔUS1R caused 60% and 50% death of animals, respectively. In contrast, no duckling died in the 2ΔUS1- and MEM-inoculated groups.

According to the results of group 1, we found that the peak of duck mortality was between 4 and 7 days postinoculation. To better evaluate the variation in DPV virulence *in vivo* after US1 deletion, 24 14-day-old ducklings were inoculated with MEM or $10^6$ 50% tissue culture infective doses ($TCID_{50}$) of CHv, 2ΔUS1, or 2ΔUS1R. At 5 days postinoculation (dpi), target organs (liver, spleen, thymus, bursa, and duodenum) were collected for evaluation of gross and histopathological lesions. As expected, the organs in the control group were healthy without any visible lesions, while all the collected organs showed obvious pathological findings, such as bleeding and enlargement, after infection with the wild-type strain (CHv) and revertant virus (2ΔUS1R). In contrast, symptoms after infection with the US1 deletion virus (2ΔUS1) were relatively mild, with only mild liver swelling and thymus bleeding observed (Fig. 5E). The collected liver, spleen, and duodenum of each group were paraffin sectioned and stained with hematoxylin and eosin (HE) to observe microscopic lesions. As shown in Fig. 5F, no microscopic lesions were observed in the control group (mock), while mild microscopic lesions of tissues were observed in ducklings inoculated with the US1 deletion mutant. After 5 days postinoculation, the 2ΔUS1 group showed mild inflammatory cell infiltration in the portal area of the duck liver and hyperemia in the red pulp of the duck spleen, but no significant lesions were observed in the duodenum. Compared to the control and 2ΔUS1 groups, inoculation with CHv and 2ΔUS1R resulted in severe tissue lesions. After inoculation with the same dose of CHv and 2ΔUS1R, diffuse hepatocyte edema, vacuolar degeneration, partial hepatocyte necrosis, and disordered liver structure were observed in the duck liver. Meanwhile, pathological observation in the spleen showed that the number of cells was decreased and vacuolated, and inflammatory cell and macrophage infiltrations were observed in the red pulp of the spleen. Observation of the duodenum showed mild necrosis in the mucosal epithelium. Taken together, these results indicated that the virulence of CHv lacking the US1 gene was significantly attenuated in ducks, but the virus still retained a certain pathogenicity.

**Effects of the US1 gene on viral load and gene transcription in duck tissues.** To determine whether US1 plays a role in CHv replication and pathogenesis *in vivo*, we infected ducklings with $10^6$ $TCID_{50}$ of CHv, 2ΔUS1, or 2ΔUS1R. The duodenums, livers, and spleens of infected ducklings were collected on the 5th day postinfection, and the numbers of CHv genome copies were determined by TaqMan qPCR probes for the UL30 gene. As a result, we found that the viral genome copy numbers in the selected organs of CHv- and 2ΔUS1R-infected ducklings showed no difference, while the viral loads of 2ΔUS1 in the liver and duodenum were significantly lower than those of CHv and 2ΔUS1R. Surprisingly, we found no significant difference in viral loads in the spleen among the three groups (Fig. 6A). We attributed this to the special nature of the immune organ, but this needs to be confirmed by more tests. The detection result of viral shedding ability after 2ΔUS1 inoculation showed reduced copy numbers in cloacal samples (Fig. 6B), suggesting lower shedding of virus. Next, real-time quantitative

**FIG 5** Legend (Continued)

negative control. Group 1 (*n* = 40, 10 ducklings per group) was used for measuring body weight, temperature, and survival rate for 10 days, and group 2 (*n* = 24, 6 ducklings per group) was sacrificed at 5 dpi to detect lesions, viral loads, and gene transcription. (B to D) Temperature (B), body weight (C), and survival curves (D) of ducks inoculated with the indicated viruses. Significance was determined by Mantel-Cox log rank tests (*, $P < 0.05$). (E) Gross lesions of the indicated virus-infected duckling organs. (F) Histopathological variations in the inoculated ducklings at 5 days postinfection. All images were captured at the same magnification ($\times100$), and representative images are presented.

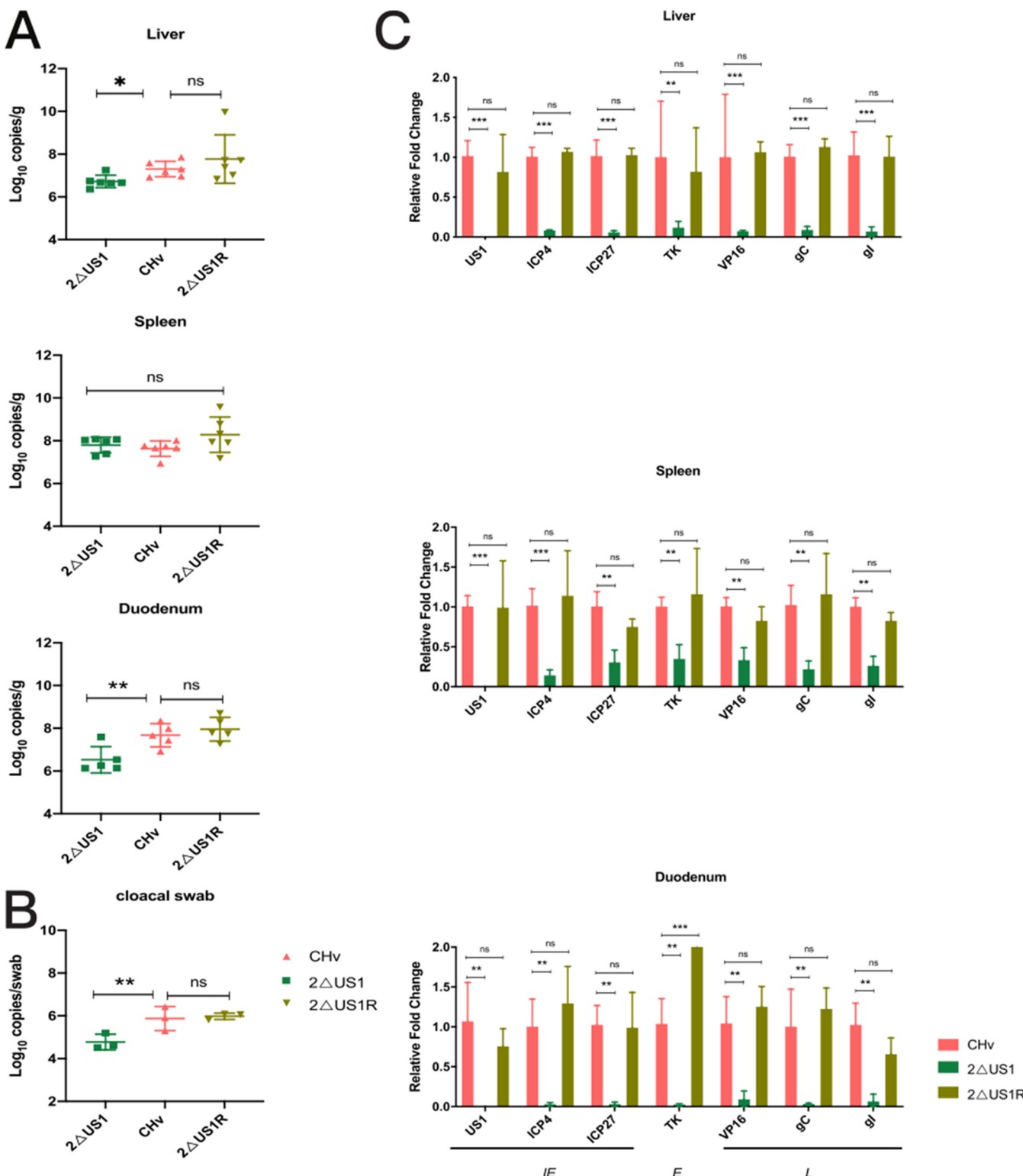

**FIG 6** Viral replication and gene transcription in duck tissues. (A) Viral genome copy numbers in the tissues of six infected ducks per group at 5 dpi measured by genomic DNA qPCR probing for the viral polymerase gene UL30. (B) Viral shedding from the cloacae of ducks ($n = 3$) at 5 dpi. (C) Relative mRNA expression levels of viral genes in tissues of six infected ducks per group at 5 dpi. The results represent mean values with error bars showing the standard error of the mean. Comparisons between CHv and 2ΔUS1/2ΔUS1R were analyzed using one-way ANOVA (A and B) or two-way ANOVA (C) and were considered significant as follows: *, $P < 0.05$; **, $P < 0.01$; ***, $P < 0.001$; ****, $P < 0.0001$; ns, not significant.

PCR (RT-qPCR) was performed on a panel of representative genes to determine the effect of the US1 gene on viral gene transcription *in vivo*. As expected, compared to CHv, the loss of US1 genes significantly reduced the transcription levels of various stages of DPV genes in the liver, spleen, and duodenum at 5 dpi, which was in accordance with the *in vitro* experimental results. In contrast, no differences were detected between CHv and 2ΔUS1R (Fig. 6C).

## DISCUSSION

DPV is one of the most harmful pathogens to the waterfowl industry (2, 40). Understanding the functions of its encoded proteins and the involved pathogenesis is one of the prerequisites for developing new vaccines. The DPV genome is large, and the functions of a large number of genes urgently need to be decoded. As it is a recently identified immediate early gene of DPV, the functions of US1 in viral replication and pathogenesis remain unknown. Here, we constructed a double US1 gene deletion mutant (2ΔUS1) and its revertant (2ΔUS1R) on the basis of the BAC-2ΔUS1 strain by using the RED/ET method (7, 39, 41, 42). Then, we completely removed the mini-F cassette from the DPV genome by intracellular homologous recombination (43) for *in vivo* experiments. In an earlier study, the thymidine kinase (TK) gene was selected as the insertion site of BAC for constructing a DPV rescue system platform (DB), which can be used to manipulate the DPV genome (39). Further characterization of DB *in vitro* and *in vivo* found that the insertion of the mini-F cassette in the TK gene did not affect the proliferation ability of CHv *in vitro*, but no genome copy numbers of DPV could be detected in DB-inoculated ducklings (unpublished data). We speculated that there are two possible reasons for this result. First, although the conserved TK gene in herpesvirus is nonessential for virus replication (44–46), it is reported to be an important virulence gene in some herpesviruses (47–50), and the insertion of the mini-F cassette resulted in the production of truncated TK protein, which would likely impair the virulence or replication of DPV. Second, the introduction of bacterial sequences may also result in foreign sequence scars, for example, the mini-F cassette, the impact of which on viral replication is unpredictable. Reports have indicated that the residue of mini-F sequences can have negative effects on virus replication: for example, they might challenge the packaging capacity of the herpesvirus capsid (51).

Therefore, we applied an inversely oriented sequence duplication between the selection marker (antibiotic resistance gene) and the bacterial replicon that allows stable maintenance of the BAC-2ΔUS1 genome in *E. coli*, which could not affect viral manipulation in the future but could trigger a self-excision of the mini-F cassette from BAC-2ΔUS1 upon transfection. As shown in Fig. 2B, five passages in cultured cells were required to completely remove the mini-F cassette from the 2ΔUS1 and 2ΔUS1R genomes, as plaques completely devoid of green fluorescence appeared at P5. One possible explanation for this phenomenon could be the cell-associated nature and slow replication kinetics of DPV, which allow the maintenance of mini-F-containing and mini-F-removed genomes in the first four cell culture passages. In addition, the recombinant virus carries a mini-F cassette lacking a growth advantage in host cells, resulting in the gradual dominance of recombinant viruses with a genomic structure more similar to that of wild-type strains following multiple rounds of infection. On this basis, we speculated that the removal of the mini-F cassette may be accelerated, especially when it is present in genes that are important for viral growth and replication. Genomic- and protein-level confirmations of 2ΔUS1 and 2ΔUS1R ensured the deletion and revertant of the US1 gene in DPV. The construction of 2ΔUS1 and 2ΔUS1R in our study allowed the self-excision and removal of the mini-F cassette from the DPV genome. It provided material for the study of US1 function *in vitro* and *in vivo*, and it also laid a foundation for the potential evaluation of the 2ΔUS1 strain as a vaccine since the residual bacterial sequences are often unfavorable for the development and licensing of live-attenuated vaccines (52).

The herpesvirus replication cycle includes attachment, fusion, entry, uncoating of viral DNA, gene transcription, genome replication, protein synthesis, packaging,

envelopment, and release (53), and influences on any of the above-described steps could result in a change in the titer of the virus. In this paper, we found the production of progeny virus in 2ΔUS1 was about 6 to 32 times less than that in CHv, while the decline in genome copy numbers of 2ΔUS1 was relatively modest, about 1.5 to 4 times lower than that of CHv. This may be related to the involvement of US1 in multiple steps of the viral replication cycle, such as packaging, envelopment, or release. A good proof of the above conjecture exists in another electron microscopic study of BAC-2ΔUS1, through which we found that US1 gene deletion could decrease the production of mature virions by inhibiting their nuclear budding and assembly (unpublished data). In addition, the number of plaques formed by 2ΔUS1 in DEFs was approximately 62% of that formed by 2ΔUS1R and CHv by plaque assay, which may be attributed to the effect of US1 deletion on virus cell-to-cell transmission because of the cell-associated nature of DPV. In addition to that, another factor that must be considered for the significant decrease in virus production after US1 deletion may be the decreased expression levels of US10 and US8 genes. Although PCR and sequencing results suggest no unexpected mutations in adjacent genes US8 and US10 caused by US1 deletion, the mRNA expression levels of US10 and US8 were significantly reduced at an early stage of 2ΔUS1 infection. This may be related to the transcriptional regulation function of the US1 gene, or it may be caused by the deletion of the US1 gene affecting the promoter of US8 and US10 genes. Given the important roles of US10 and US8 genes in viral virulence and cell-to-cell transmission (2, 54), it is difficult to say whether the decrease in viral titer after US1 gene deletion is directly caused by US1 gene deletion or indirectly caused by decreased expression levels of US8 and US10 genes. But from the point of view of vaccine development, it is certainly beneficial in terms of improving the safety of the recombinant virus.

To further study the roles of US1 in DPV gene expression in DEF cells, we selected the expression of representative genes expressed by the virus in different phases; the IE genes selected were ICP4 and ICP27, the E gene selected was TK, and the L genes selected were VP16, gI, and gC. The results showed that the transcriptional levels of the tested genes were significantly reduced after US1 deletion, indicating that the US1 gene promotes the transcription of viral genes *in vitro*. The US10 and US8 complementary experiment confirmed the transcriptional regulation function of the US1 gene on viral gene expression (data not show). The findings of US1 effect on gene transcription are consistent with those of HSV-1 US1, which could promote the expression of the L gene (55), and HSV-1 VP16 removes the transcriptional inhibition by US1 of the IE gene, inhibits cell gene expression, and promotes viral gene expression. We presumed that DPV US1 may function as HSV-1 ICP22 in gene transcription, but this hypothesis needs further verification. The above-described experimental results indicated that the double copy deletion of the US1 gene reduces the replication of DPV *in vitro*.

Although the US1 gene can be carried by all alphaherpesviruses, its pathogenesis is not the same among different viruses. As it is one of the two duplicated genes in the DPV genome, the function of US1 in DPV pathogenesis is still unknown. We first tested the roles of US1 in the virulence and pathogenicity of DPV *in vivo*. The results showed that 2ΔUS1 showed attenuated virulence and that the inoculated ducklings had significantly reduced gross lesion development and mortality during the observation period, but 2ΔUS1 was still mildly pathogenic to the ducklings. This was reflected in the 100% survival rate after 2ΔUS1 inoculation, but the body temperature of the ducks increased significantly to more than 43°C within 3 to 5 days postinfection. The weight gain of the ducklings slowed compared with the control, with slight macroscopic lesions of the liver and thymus and microscopic lesions of the liver and spleen. Further detection of viral loads and shedding indicated that viral genome copy numbers in the liver, duodenum, and cloacal swabs of infected ducklings were significantly lower than those of wild-type and revertant viruses at 5 dpi. Therefore, we hypothesized that 2ΔUS1 leads to the reduced replication ability of DPV in ducklings or its colonization ability in certain tissues and organs, making it easier for DPV to be cleared by the host immune system. Surprisingly,

we found that the spleen was an exception because there was no significant difference in viral loads between 2ΔUS1, 2ΔUS1R, and CHv, and the viral loads of all three groups in the spleen were maintained at high levels. We attributed this to the specific nature of the immune organs, but this has not been confirmed because we did not measure viral loads in other immune organs. Another hypothesis is that DPV mainly attacks the immune organs of ducklings after infection, which results in serious damage to the immune organs of the body and reduced virus-clearance ability. This could also explain why ducks infected with the 2ΔUS1 strain survived but still displayed increased body temperature, decreased body weight, and mild tissue damage. However, more rigorous experiments are needed to confirm this possibility.

Among other herpesviruses, US1 is best known and recognized for its transcriptional regulation of viral and host genes (56–58). To study the effects of DPV US1 on viral gene transcription and virulence, we selected some representative viral genes for mRNA-level detection. VP16 is the transcriptional activator of herpesvirus, which can assist the US1 protein in regulating the transcription of viral genes (59–62). ICP4 is the only transcription factor of herpesvirus that has been reported thus far and can activate the transcription of genes at various stages of the virus but inhibit its own transcription (63–65). ICP27 is conserved in all herpesviruses, and its most important function is to participate in the transport, splicing, and modification of viral mRNA and inhibit the splicing of host genes, which is very important for the efficient transcription of viral genes (66, 67). TK, gC, and gI are closely related to herpesvirus virulence. TK encodes thymidine kinase, which is an important virulence gene product of herpesvirus that can affect the lytic and latent periods of the virus infection. gC and gI are virus-encoded glycoproteins that affect virulence by affecting the adsorption and intercellular transmission of herpesvirus, respectively (68–71). The above-described tested genes are highly conserved in herpesviruses and are associated with the transcriptional process and virulence of viral genes. They can also be used as typical representatives of the regulatory ability of US1 genes in different periods. *In vitro* and *in vivo* analyses of 2ΔUS1 effect on viral gene transcription were in accordance, and the results showed decreased mRNA levels of IE (ICP4 and IPC27), E (TK), and L (VP16, gC, and gI) in 2ΔUS1-infected DEF cells and ducklings compared to those in ducklings infected with CHv and 2ΔUS1R. Therefore, we speculated that US1 can regulate viral gene expression in all stages, which was in accordance with US1 homologs in other herpesviruses. The decreased expression levels of virulence genes (TK, gC, and gI) after US1 deletion could be one of the possible explanations for the attenuated virulence of 2ΔUS1. However, more experiments are needed to verify the above hypothesis and investigate the underlying mechanism.

**Conclusions.** In conclusion, we generated self-excisable BAC clones of DPV 2ΔUS1 and 2ΔUS1R by employing a stabilized inverse sequence duplication inside the mini-F cassette and reconstituted 2ΔUS1 and 2ΔUS1R viruses upon transfection of DEF cells. Characterization of the US1 deletion mutant suggested that loss of the US1 gene reduces viral gene expression, replication, and virus production both *in vitro* and *in vivo*, and 2ΔUS1 was greatly attenuated in ducklings, which indicated that US1 is involved in the regulation of virus replication and pathogenesis. These findings provide clues for the study of the pathogenesis and development of attenuated vaccines targeting the US1 gene.

## MATERIALS AND METHODS

**Viruses and plasmids.** The duck plague virus Chinese virulent strain (CHv) was used as the wild-type virus. The BAC-2ΔUS1 plasmid maintained in *E. coli* was constructed earlier by our group and used as the parental plasmid for generating the 2ΔUS1 mutant without mini-F cassette retention.

**Cells.** Duck embryo fibroblast (DEF) cells were prepared from 9- to 11-day-old healthy duck embryos (Chengdu Kerimo Breeding Company, China) as previously described (7). Cells were maintained in Dulbecco's modified Eagle's medium (DMEM) supplemented with 10% newborn bovine serum, 250 U/mL penicillin, and 250 $\mu$g/mL streptomycin in a 37°C humidified incubator under a 5% $CO_2$ atmosphere.

**Construction and rescue of the 2ΔUS1 and 2ΔUS1R mutant viruses.** In our previous study, we generated double copies of a US1 deletion mutant with a mini-F element (BAC-2ΔUS1) through a two-step Red recombination system of DPV CHv (41, 43). The two copies of the US1 gene were deleted one

**TABLE 1** Primers and probes used in this study

| Primer | DNA sequence (5′–3′) |
| --- | --- |
| US1-F | TCATTGCTCAATACGGGAAG |
| US1-R | GCGGTGTTTATTGACATCA |
| UL30-F | TTTTCCTCCTCCTCGCTGAGT |
| UL30-R | GGCCGGGTTTGCAGAAGT |
| UL30-probe | FAM*a*-CCCTGGGTACAAGCG-MGB |
| ICP4-F | AATCTATGCCCGTCCAAGCTC |
| ICP27-R | CCCGGACCCATTACTAGGCACA |
| ICP27-F | ACCTACAATTCAGCAACGCATA |
| TK-F | CCACCAGATATTACGCTCA |
| TK-R | CCAATAGAGTACTAAGGCTCA |
| gI-F | TCTTGGATCACAGGCCGAAC |
| gI-R | AGCTGCATACGCGACAGAAT |
| gC-F | CGAATCATAAAGGGCCGCATC |
| gC-R | ATTAGATCTCGTTACCCGCTTG |
| US8-F | GACAGCTCTGGGGATCAT |
| US8-R | CGTATTGACCTTTTGCGA |
| US10-F | CGTATTGACCTTTTGCGA |
| US10-R | ATATGTACACGACACCGC |
| BAC-KanR-F | TTATTAATCTCAGGAGCCTGTGTAGCGTTTATAGGAAGTAGTGTTCTGTCATGATGCCTGCAAGCGGTAACGAAAACG ATTGTTACAACCAATTAACC |
| BAC-KanR-R | ATCGTTTTCGTTACCGCTTGCAGGCATCATGACAGAACACTACTTCCTATTAGGGATAACAGGGTAATCGAT |
| BAC-UL23-F | GCCTGCAAGCGGTAACGAAAACGATTCAATTAATTGTCATCTCGG |
| BAC-UL23-R | CCGCTCCACTTCAACGTAACACCGCACGAAGATTTCTATTGTTCCTGAAGGCATATTCAACGGACATATTAAAAATTGA |

*a*FAM, 6-carboxyfluorescein.

by one, and we obtained 2 subsequent recombinations, named BAC-ΔUS1 and BAC-2ΔUS1. The construction details have been described in our previous studies. However, the major disadvantage of this mutant was the presence of a mini-F cassette in the TK region. Although the presence of the mini-F cassette in the TK gene did not affect the proliferation of the virus on cells in our previous studies, whether the insertion of the mini-F cassette in the TK region produces an effect on the virus *in vivo* was not guaranteed since TK is known to be important for the virulence of herpesvirus, and no reports of the mini-F cassette effect on virulence have been made yet. Therefore, to define the role of the US1 gene more accurately in the pathogenicity of duck plague virus *in vivo*, we constructed 2ΔUS1 and its revertant that did not contain any other mutations except the US1 gene on the basis of BAC-2ΔUS1 according to previous reports (43). The schematic diagrams of 2ΔUS1 and 2ΔUS1R construction are shown in Fig. 1. Briefly, a 2.7-kb linear DNA fragment containing the Kana-I-SceI cassette (amplified from pEPKan-S) and the TK sequence-flanked mini-F element (amplified from DEV CHv) were introduced into the bacterial origin of replication (ori2) and CmR of the mini-F element of BAC-2ΔUS1 by the first round of Red recombination. The resulting colonies were selected by 30 μg/mL kanamycin and 30 μg/mL chloramphenicol on LB agar plates. The second round of Red recombination was triggered by 1% L-arabinose to remove the Kana-I-SceI cassette. PCR identification and sequencing were performed to confirm the obtained colonies, and the correct colonies were named p2ΔUS1. To exclude the influence of nontargeted mutations caused by genetic engineering operations, the revertant mutant p2ΔUS1R was constructed based on p2ΔUS1 by Red recombination, and the correctness of these mutated plasmids was identified by PCR and sequencing.

The plasmid p2ΔUS1 and p2ΔUS1R DNAs were extracted and transfected into DEFs for virus reconstitution, and plasmid isolation and transfection were performed following the manufacturer's instructions for the Qiagen plasmid midikit and Lipofectamine 3000 (Invitrogen, USA). A schematic diagram of the rescue, mini-F self-excision, and purification of the recombinant virus is shown in Fig. 2A. As shown, the successfully rescued virus displayed green plaques after transfection, and the green fluorescence gradually disappeared during virus passage via intra- and intermolecular recombination events due to the duplicated sequence around and inside the mini-F cassette. Usually, plaques without any fluorescence will be obtained after 3 to 4 passages, which indicates that the mini-F cassette is completely removed from the virus genome. It is worth noting that limited dilutions and pickup of the fluorescence-free plaques can accelerate the purification of mutants. PCR identification and sequencing were performed to confirm the deletion and revertant of the US1 gene in the rescued recombinant virus and the genomic stability of its neighboring genes US10 and US8. Meanwhile, the protein level of the US1 gene was identified by WB and IFA, and the viral protein VP16 was also detected as an indicator of viral infection. The obtained recombinant viruses confirmed by PCR, sequencing, WB, and IFA were designated 2ΔUS1 and 2ΔUS1R. All primers used for mutagenesis and identification of 2ΔUS1 and 2ΔUS1R are listed in Table 1.

**Western blotting.** For Western blotting of US1 mutants, 2ΔUS1, 2ΔUS1R, or wild-type CHv at an MOI of 2 was inoculated into DEF cells grown in 12-well plates. Cells were harvested at 48 h postinfection (hpi) and centrifuged at 10,000 rpm for 5 min, and mock-infected cells were used as a negative control. After the supernatant was discarded, 100 μL radioimmunoprecipitation assay buffer (RIPA) and 1 μL

phenylmethylsulfonyl fluoride (PMSF) were added to the cell pellets to dissolve the cell precipitate. The samples were boiled for 10 min, and the supernatants were loaded and separated by 10% SDS-PAGE. Next, proteins were transferred to polyvinylidene difluoride (PVDF) membranes (Bio-Rad, USA) by half-dry transfer at 80 V for 45 min. The PVDF membranes were incubated at 4°C overnight with 5% skimmed milk powder for blocking, which was followed by incubation with 1% bovine serum albumin (BSA)-diluted mouse anti-US1 antibody (1:400), mouse anti-VP16 antibody (1:800), or mouse anti-beta-actin antibody (1:5,000) at 4°C for one night. After that, three washes of the membranes with Tris-buffered saline with Tween 20 (TBST) were performed before horseradish peroxidase (HRP)-labeled secondary antibody (1:5,000) incubation. The bands were finally visualized by an enhanced chemiluminescence (ECL) chromogenic kit (Bio-Rad).

**Indirect immunofluorescence.** The DEFs grown on glass coverslips plated in 6-well dishes were infected with an MOI of 0.01 of CHv, 2ΔUS1, or 2ΔUS1R for IFA identification of recombinant viruses as previously described, and mock-infected cells were used as a negative control. (41). In simple terms, 1 mL of 4% paraformaldehyde was added to each well at 48 hpi to fix the cells at 4°C overnight, and then cells were washed three times with phosphate-buffered saline with Tween 20 (PBST) followed by the addition of 0.25% Triton-100 for 30 min of permeabilization. Then, the cells were covered with blocking buffer (5% BSA) at 4°C overnight before incubation with mouse anti-US1 antibody (1:200) and rabbit anti-VP16 antibody (1:200). Next, the cells were washed three times with PBST and incubated with tetramethyl rhodamine isocyanate (TRITC)-HRP-labeled goat anti-mouse/rabbit IgG antibody (1:1,000) at 37°C for 2 h. Finally, 1 $\mu$L 4',6-diamidino-2-phenylindole (DAPI) was used to visualize nuclei and incubated at room temperature for 30 min after washing with PBST. Images were observed and captured under a fluorescence microscope (Nikon, Japan).

**Growth curve and plaque formation assessment.** Growth curves and plaque size assessments were used to characterize the in vitro replication properties of the recombinant virus. For growth curve analysis, DEF cells were infected at an MOI of 0.01 (multistep assay) or at an MOI of 2 (single-step assay) in 12-well dishes. After the virus was adsorbed for 2 h at 37°C and 5% $CO_2$, the medium was discarded, the cells were washed with PBS (pH 7.4), and then the culture medium was replaced with MEM containing 2% newborn calf serum (NBS). At the indicated time points at successive intervals, infected DEFs were harvested, and serial dilutions were cocultured with fresh DEFs in 96-well plates. Each dilution was repeated for eight wells. After 7 to 10 days of infection, the number of cytopathic wells was recorded. The $TCID_{50}$ values of these viruses were calculated by the Reed-Muench method (43), and the titer growth curve of the virus was plotted. Plaque formation assessment was performed according to the literature (72). Briefly, DEF cells were infected with an MOI of 0.0001 of CHv, 2ΔUS1, or 2ΔUS1R for 2 h and then covered with methoxy methanol semisolid medium. After 7 days, the formation of plaques was observed by 4% paraformaldehyde combined with crystal violet staining. The numbers of each virus were recorded, and the areas of all plaques for each titrated virus were measured by ImageJ to calculate the diameters.

**Animal experiment.** Sixty-four day-old healthy Cherry Valley ducklings were purchased from a farm operated by Sichuan Agricultural University (Sichuan, China). No experimental ducklings were infected with DPV, and all were negative for DPV antibodies. The experimental animal protocol was approved by the Ethics and Animal Welfare Committee of Sichuan Agricultural University and was carried out following the Chinese version of the Guide for the Care and Use of Laboratory Animals. To assess the effects of the US1 gene on DPV replication and virulence in vivo, 40 14-day-old ducklings were randomly divided into 4 groups (10 ducklings per group) and intramuscularly infected with $10^6$ $TCID_{50}$ of CHv, 2ΔUS1, or 2ΔUS1R virus and the same volume of MEM. The body temperature, body weight, and mortality of each group were measured daily for 10 days. In another experiment, 24 14-day-old ducklings were randomly divided into 4 groups (6 ducklings per group) and injected with the same dose of CHv, 2ΔUS1, or 2ΔUS1R virus and MEM for sacrifice at 5 days postinfection. The tissues of each duckling were collected to analyze the lesions, viral loads, and gene expression. The swab samples from 3 ducklings in each group were collected before sacrifice for viral shedding determination.

**DPV genome copy number quantification.** To quantify the copy number of the DPV genome, DPV DNA from CHv-, 2ΔUS1-, or 2ΔUS1R-infected cells or duckling tissues was first isolated. To study the copy number of the viral genome in DEF cells, nearly confluent DEF cell monolayers in 12-well plates were infected with each virus in triplicate at either a low MOI (0.01) or a high MOI (2) and then harvested at the indicated time points postinfection. DPV DNA from infected DEF cells was subsequently extracted by a HiPure viral DNA kit (catalog no. D3191; Magen) according to the manufacturer's instructions. The viral DNA in swabs was also extracted with the same kit. To extract viral DNA from infected animal tissues, 100-mg tissue samples were first frozen in liquid nitrogen and then ground into powder with grinding steel balls before isolation by DNAiso reagent (9770A; TaKaRa). Determination of the DPV genomic copy numbers in the above-described samples by TaqMan qRT-PCR was then performed as previously described by using probes and primers specific to the UL30 gene and calculated by the previously generated standard curve: $Y = -4.262X + 43.675$ (73).

**Quantification of mRNA levels in DEF cells and duck tissues.** The relative gene expression of DPV genes in intracellular samples and tissues was determined by reverse transcription-quantitative PCR (RT-qPCR) as previously described (73). For in vitro infection, DEFs were infected with CHv, 2ΔUS1, or 2ΔUS1R virus at an MOI of 2 in 12-well plates. After adsorption at 37°C for 2 h, samples were harvested at 6 h, 24 h, and 36 h after infection (three biological replicates per group). For extraction of RNA from organs, 100 mg of tissue frozen in liquid nitrogen was first ground into powder with grinding steel balls. Then, the total RNA in cells and tissues was extracted using RNA-easy isolation reagent (catalog no. R701; Vazyme) according to the manufacturer's instructions. The extracted RNA was treated with DNase I before being reverse transcribed into cDNA by a PrimeScript real-time (RT) reagent kit (catalog no.

RR037A; TaKaRa). The relative mRNA levels of ICP22, ICP4, ICP27, TK, VP16, gI, and gC were detected using a SYBR Premix *Ex Taq* II kit (catalog no. R701; TaKaRa) with primers specific to the viral genes listed in Table 1. Briefly, 1 $\mu$L of the RT mixture was used in 10-$\mu$L volumes containing 0.4 $\mu$L forward and reverse primers. The relative amounts of viral genes were normalized to cellular 18S RNA. For fold change of gene expression, threshold cycle ($2^{-\Delta\Delta CT}$) was used to calculate the relative amounts of viral genes compared to the expression of CHv-infected samples.

**Statistical analyses.** All statistics were performed using GraphPad Prism (version 8.2.0). The data of cell and animal experiments represent the results of at least three independent experiments. Both 2ΔUS1 and 2ΔUS1R were compared with CHv, and the significance for experiments was determined by one- or two-way analysis of variance (ANOVA) (*, $P < 0.05$; **, $P < 0.01$; ***, $P < 0.001$; ****, $P < 0.0001$). The TCID$_{50}$ of the virus was calculated by the Reed-Muench method, and significance for the survival curve was examined by Mantel-Cox log rank tests.

## ACKNOWLEDGMENTS

This research was supported by the Natural Science Foundation of Sichuan Province (2022NSFSC0077), the earmarked fund for China Agriculture Research System (CARS-42-17), and the Program Sichuan Veterinary Medicine and Drug Innovation Group of the China Agricultural Research System (SCCXTD-2021-18).

We have no conflicts of interest to declare.

All authors listed contributed to the completion of the article. Y.W., S.T., and Q.H. contributed to the design and writing of the article; M.W., R.J., S.C., Q.Y., D.Z., M.L., X.Z., S.Z., J.H., X.O., S.M., Q.G., D.S., and B.T. all provided ideas contributing to the structure of this article, and A.C. modified the article. All the authors reviewed and approved the final manuscript.

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
