## [Reviewer comments · Microbiology Spectrum]

Microbiology Spectrum

Deletion of double copies of the US1 gene reduces the infectivity of recombinant duck plague virus in vitro and in vivo

Ying Wu, Silun Tan, Qing He, Mingshu Wang, Shun Chen, Renyong Jia, Qiao Yang, Dekang Zhu, Mafeng Liu, Xinxin Zhao, Shaqiu Zhang, Juan Huang, Xumin Ou, Sai Mao, Qun Gao, Di Sun, Bin Tian, and Anchun Cheng

Corresponding Author(s): Anchun Cheng, Avian Disease Research Center, College of Veterinary Medicine, Sichuan Agricultural University

Review Timeline:

Submission Date:	March 29, 2022
Editorial Decision:	April 18, 2022
Revision Received:	June 9, 2022
Editorial Decision:	June 28, 2022
Revision Received:	August 12, 2022
Editorial Decision:	September 5, 2022
Revision Received:	October 21, 2022
Accepted:	October 26, 2022

Editor: Clinton Jones

Reviewer(s): The reviewers have opted to remain anonymous.

Transaction Report:

DOI: <https://doi.org/10.1128/spectrum.01140-22>

April 18, 2022

Prof. Anchun Cheng
Sichuan Agricultural University
College of Veterinary Medicine
No. 211, Huimin Road
Chengdu, Sichuan 611130
China

Re: Spectrum01140-22 (Deletion double copies of US1 gene reduce infectivity of recombinant Duck Plague Virus in vitro and in vivo)

Dear Prof. Anchun Cheng:

Thank you for submitting your manuscript to Microbiology Spectrum. While both reviewers found the manuscript to be interesting, there are many typos and grammatical errors in the manuscript that must be addressed. Furthermore, both reviewers emphasized that someone who speaks excellent English needs to assist with revising the manuscript. It is important that you use the reviewers comments as a guide for revising the manuscript. When submitting the revised version of your paper, please provide (1) point-by-point responses to the issues raised by the reviewers as file type "Response to Reviewers," not in your cover letter, and (2) a PDF file that indicates the changes from the original submission (by highlighting or underlining the changes) as file type "Marked Up Manuscript - For Review Only". Please use this link to submit your revised manuscript - we strongly recommend that you submit your paper within the next 60 days or reach out to me. Detailed instructions on submitting your revised paper are below.

Link Not Available

Sincerely,

Clinton Jones

Journals Department
Reviewer comments:

Reviewer #1 (Comments for the Author):

Duck plague virus (DPV) is an alpha-herpesvirus that can cause acute disease (duck plague) with high mortality rates in flocks of ducks, geese and swans. Many of the viral genes in DPV have not been studied regarding what viral genes are essential for replication of the virus in vivo and for pathogenesis. The goal of the study was to test the role of US1 gene of DPV in viral

replication, viral gene expression, and virus production in vitro and in vivo as well as in pathogenesis in vivo. To this end, the two copies of US1 gene were deleted from DPV and revertant was also generated. Duck embryo fibroblast and ducklings were infected and virus growth was analyzed. The authors concluded based on their results that US1 deletion reduces viral gene expression, replication and virus production both in vitro and in vivo. They also found that in contrast to WT DPV, the US1 deletion mutant virus did not kill ducklings after infection. Their data indicate that US1 is involved in the regulation of virus replication and pathogenesis. The experiments were logically designed, executed, and presented but the study in its current form raises several questions.

Comments

1. I strongly recommend that English of the manuscript be thoroughly checked for typos, grammar, words...Also, there are long sentences that need to be broken up. In many cases wrong words are used or words are missing from sentences.
2. The way the study is presented it seems as if they made the double US1 deletion mutant virus in this study despite that it has been published by the same research group 2 years ago (PMID: 32010642). This needs to be addressed. I recommend that they delete and/or re-write the mutant virus generation section addressing what is different here.
3. I am curious whether the deletion of US1 genes affects the neighboring genes US10 and US8, which can influence the interpretation of the phenotype. If US1 deletion also abrogated the expression of US10 and/or US8, they can mask US1 phenotype either reducing or increasing the effect of US1 deletion.
4. It is not clear how the duplicated copies of US1 were deleted. One after the other, that is, that there were 2 subsequent recombinations? The description of bacmid recombination in the Methods section is not understandable.
5. In the text and figures two groups of ducklings were used for DPV infection, 10 and 6 birds, respectively. In the methods there are 40 and 24 ducklings per group.
6. How the viral DNA copy number was calculated in infected cells? Was it normalized to host DNA? Using different amount of total DNA from different samples will skew the measurement of viral DNA.
7. Figure 4: The difference in viral gene expression between WT and 2deltaUS1 appears to be negligible. I do not consider them substantial changes, which would explain the few fold reduction of virus production. It is unclear what the asterisks mean above the bars. In the figure legend the text says that "differences between two groups were analyzed using Student's t test...". What two groups were compared to each other? The place of asterisk/p-value calculation is incorrect.
8. In the last figure, why different numbers of ducks were used for viral DNA measurement in different tissues according to the graphs? The viral DNA was not tested in the indicated tissues in all infected ducks?

Reviewer #2 (Comments for the Author):

The manuscript "Deletion double copies of US1 gene reduce infectivity of recombinant Duck Plague Virus in vitro and in vivo" by Wu et al. assessed the role of the US1 gene in the pathogenesis, replication, and gene expression of duck plague virus (DPV). The authors constructed two recombinant viruses, one with the deletion of the double US1 gene found in DPV, the other a revertant of this deletion. These recombinant viruses were compared in vitro and in vivo for their replication. The double deletion mutant resulted in reduced viral titers in growth kinetics, reduced number and size of plaques, and reduced copies of viral DNA in vitro. This mutant also resulted in reduced lesions and no mortality compared to the revertant and the wildtype control in ducks, with reduced titers in some organs. The manuscript provide interesting insight into the role of the US1 gene in DPV virulence, showing that its deletion results in an attenuated virus that has the potential to be used as a vaccine candidate. However, these findings are masked by the poor quality of the writing and often superficial description of details.

Major comments

The manuscript needs significant editing of the English language. Articles and prepositions are missing throughout the manuscript (e.g., the title should be "Deletion of double copies of US1 gene reduces infectivity of recombinant Duck Plague Virus in vitro and in vivo"), and many sentences are confusing. A few examples are given in the specific comments below, but authors are encouraged to have the text revised by a native speaker or editing service.

The manuscript is not very explicative to a broader audience, with many technical terms not described, and many abbreviations used without explanation (e.g., VZV, PRV, etc) or that need to be better defined, at least the most discussed ones (e.g., ICP22). Although the methods seem to be sound, they are mostly superficially described, and many details are lacking and need to be expanded. More specific comments are below. Additionally, pathological findings are described in a non-technical way, and should be revised by a pathologist to provide correct wording. In addition, some of the statistical analysis seem to be off, with (highly) significant statistical differences found for some graphs that don't seem to have such different values or have large

variations (e.g., Figs 3A, 4, 6A).

The discussion is very confusing, and mostly superficial. Not much discussion about the genes that were tested.

Specific comments

The abstract and the Importance are very contrasting - while the abstract emphasizes pathogenesis, the importance is solely focused on vaccine production. Although it is something that could result from this work, production (or testing) of a vaccine was not the point of this manuscript.

L13-14: DPV possess a large genome consisting of 78 ORFs. Understanding the function and mechanism of each protein

L15-16: US1 is one of the two genes located in the repeat regions of DPV genome, but the function of its encoding protein in DPV

L17: Previous studies - please modify throughout

L21: that deletion of both copies

L22: could represent a potential candidate vaccine

L23: remove the word occurrence

L30: Change "can overcome the deficiency of them, which will be helpful for the epidemiological" to "can be differentiated from natural infection, which will be helpful for the epidemiological"

L32: available against DPV yet

L38: alternatively

L39: causative agent

L54: but is not an IE gene

L58: which can directly interact with cyclin-dependent kinase (CDK9) and inhibit the enzyme activity

L60: inhibits

L64: in involved in virus immune evasion

L79: The DPV genome includes a unique long (UL) and unique short (US) region - this description needs to be first mentioned in the previous paragraph

L82: US1 gene coexisting. The other duplicated gene is ICP4.

L84: we generated a deletion mutant with deletion of the double US1 genes - the way this mutant is termed is often confusing throughout the manuscript. Perhaps the best way to term it is "double US1 gene deletion mutant". I suggest using the abbreviation 2 Δ US1 as often as possible.

L90: since results come first, the abbreviations should be spelled out first in the results

L94: an example of how the terms used for the mutant are not clear: "double copies of US1 gene deletion strain 2 Δ US1"

What is the difference between the wild type and the revertant virus? It would be good to have an explanation of the rationale to have both.

L107: was used to indicate

L112: These results suggest

L124-125: wild type and revertant strains at every time point (Fig.3B). Plaque assays were performed to evaluate

L126: US1 gene resulted in smaller

L131: "indicating US1 gene promotes the transcription of DPV gene in all phases" - this is confusing. Which genes? Which phases?

L134: viral pathogenesis, and the flow chart of the animal experiment is shown in

L141: died

L145L post-inoculation

L146L collected and evaluated for gross and histopathological lesions.

L147: without any visible (or noticeable) lesions

L160: Livers of birds infected with

L161: Histopathological analysis of the spleen showed

L171: while DNA quantification of 2 Δ US1 - measuring of genetic material does not necessarily correlate to replication, this was only measured at one time point, so there is no way to know if it increased.

L174: The deletion of US1 gene reduced the shedding rate

L188-189: An detailed explanation of the mini-F cassette should have been mentioned in the methods section, not only here.

L194: resulting in changes

L196: restore TK function to study the role

L200: viruses are purified

L206: by hiding

L212: tested

L217: but this hypothesis needs further verification

L224-225: had significantly reduced gross lesion

L226: completely avirulent

L236-238: This sentence does not relate much with what was found in the spleen, it should be placed before that sentence about the spleen.

L245-246: This sentence is not very coherent. Rephrase it.

L249: viral gene transcription in each tissue: This was true even for the spleen, despite no differences in titers. That is something

that needs to be discussed.

L332: more details about animal experiments are needed - how were animals maintained, N, info about IACUC, species, etc.

L337: was MEM administered to the Mock control?

L338: When were swabs and tissues collected and what tissues?

L341: from cells or tissues were isolated.

L357-358: remove the repeated sentence: For extractions of RNA in organs, tissues frozen in liquid nitrogen should be ground into powder with grinding steel balls at first.

Staff Comments:

Preparing Revision Guidelines

Please return the manuscript within 60 days; if you cannot complete the modification within this time period, please contact me. If you do not wish to modify the manuscript and prefer to submit it to another journal, please notify me of your decision immediately so that the manuscript may be formally withdrawn from consideration by Microbiology Spectrum.

Review 1#:

Duck plague virus (DPV) is an alpha-herpesvirus that can cause acute disease (duck plague) with high mortality rates in flocks of ducks, geese and swans. Many of the viral genes in DPV have not been studied regarding what viral genes are essential for replication of the virus in vivo and for pathogenesis. The goal of the study was to test the role of US1 gene of DPV in viral replication, viral gene expression, and virus production in vitro and in vivo as well as in pathogenesis in vivo. To this end, the two copies of US1 gene were deleted from DPV and revertant was also generated. Duck embryo fibroblast and ducklings were infected and virus growth was analyzed. The authors concluded based on their results that US1 deletion reduces viral gene expression, replication and virus production both in vitro and in vivo. They also found that in contrast to WT DPV, the US1 deletion mutant virus did not kill ducklings after infection. Their data indicate that US1 is involved in the regulation of virus replication and pathogenesis. The experiments were logically designed, executed, and presented but the study in its current form raises several questions.

The author's response:

We thank the reviewer for reading our paper carefully and giving the above positive comments. We have carefully made a comprehensive revision point-by-point in accordance with every comment. Our description on revision according to the comments as follows. Meanwhile, we resubmitted our manuscript with tracked changes that are highlighted in red.

Q1: strongly recommend that English of the manuscript be thoroughly checked for typos, grammar, words...Also, there are long sentences that need to be broken up. In many cases wrong words are used or words are missing from sentences.

The author's response:

Thanks for the kind comments and suggestions, they are very valuable in improving the quality of our manuscript. We attached great importance to every comment made by you. Therefore, we requested a professional service of English proofreading from American Journal Experts to revise the writing problems of this article. The other grammar and statement errors have been carefully corrected in the revised manuscript, which are highlighted. Meanwhile, we resubmitted our manuscript with tracked changes that are highlighted in red. We hope the revised manuscript can meet the language requirements of Microbiology Spectrum and you can consider giving this manuscript the opportunity to be published in this journal.

Q2: The way the study is presented it seems as if they made the double US1 deletion mutant virus in this study despite that it has been published by the same research group 2 years ago (PMID: 32010642). This needs to be addressed. I recommend

that they delete and/or re-write the mutant virus generation section addressing what is different here.

The author's response:

Thank you very much for your insightful comments and valuable suggestions. As you mentioned, we have constructed a DB-2 Δ US1 mutant 2 years ago (PMID: 32010642). However, the major disadvantage of this mutant was the presence of mini-F cassette in TK region. Although the presence of Mini-F cassette in TK gene did not affect the proliferation of the virus on cells in our previously studies, the insertion of Mini-F in TK region on the virus in vivo was not guaranteed, since TK is known to be important for the virulence of herpes virus and no reports of mini-F cassette on virulence have been reported yet. Therefore, to more accurately define the role of US1 gene in the pathogenicity of duck plague virus in vivo, we constructed the 2 Δ US1 and its revertant that did not contain any other mutations except US1 gene on the basis of DB-2 Δ US1(as shown in Figure 1, line 3). In consideration of your comments, we have rewritten the mutant virus generation section and addressed the differences between DB-2 Δ US1 and 2 Δ US1 in materials, please review the highlighted content in the resubmitted manuscript(line 350-353).

Q3: I am curious whether the deletion of US1 genes affects the neighboring genes US10 and US8, which can influence the interpretation of the phenotype. If US1 deletion also abrogated the expression of US10 and/or US8, they can mask US1 phenotype either reducing or increasing the effect of US1 deletion.

The author's response:

Thanks very much for your valuable suggestions. According to your suggestion, we tested the integrity of the neighboring genes US10 and US8 by PCR and sequencing. The results were supplemented in resubmitted Figure 2C and corresponding result section. According to the result, we believe that the deletion of US1 gene has no effect on the stability of adjacent genes since no mutations of US10 and US8 gene have been found by PCR and sequencing, which can also indicate that the phenotype variations are only caused by US1 deletion. Please review the highlighted content in the resubmitted manuscript(line 119-123).

Q4: It is not clear how the duplicated copies of US1 were deleted. One after the other, that is, that there were 2 subsequent recombinations? The description of bacmid recombination in the Methods section is not understandable.

The author's response:

Thank you very much for your insightful comments. The two copies of US1 gene were deleted one by one, and we have obtained 2 subsequent recombinations named BAC- Δ US1 and BAC-2 Δ US1, the construction details have been described in our previously studies yet(PMID: 32010642). Meanwhile, the focus of this paper is to remove the miniF cassette in DB- Δ 2US1 and restore the US1 gene. Therefore, the

deletion of US1 gene has not been described in detail in the method section in this paper. According to your suggestion, we have added a more detailed interpretation regarding the deletion of US1 genes. Please review lines 341-345 and lines 353-360.

Q5: In the text and figures two groups of ducklings were used for DPV infection, 10 and 6 birds, respectively. In the methods there are 40 and 24 ducklings per group.

The author's response:

Thank you so much for your careful check. We are sorry for the inappropriate statement. We have re-described the grouping of ducks in animal experiments in the method section and made corresponding modifications in Figure 2. Please review lines 425-433.

Q6: How the viral DNA copy number was calculated in infected cells? Was it normalized to host DNA? Using different amount of total DNA from different samples will skew the measurement of viral DNA.

The author's response:

Thank you for pointing out this problem of our manuscript. We are sorry for the inappropriate statement. In this paper, the same amount of infected cells or tissues were used for DPV DNA isolation, and the quantification of viral DNA were determined by the TaqMan qRT-PCR which has been established previously in our lab. The accurate copies of DPV DNA was calculated by the standard curve: $Y = -4.262X + 43.675$. Therefore, there was no need to normalize it to host DNA. To be more clearly and in accordance with the reviewer concerns, we have added more detailed interpretations regarding DNA copy number determination method. Please review lines 440-445 of the resubmitted manuscript.

Q7: Figure 4: The difference in viral gene expression between WT and 2deltaUS1 appears to be negligible. I do not consider them substantial changes, which would explain the few fold reduction of virus production. It is unclear what the asterisks mean above the bars. In the figure legend the text says that "differences between two groups were analyzed using Student's t test...". What two groups were compared to each other? The place of asterisk/p-value calculation is incorrect.

The author's response:

Thanks for the kind comments and suggestions. Considering our experimental data had at least three groups and other variables were involved, we used One-way and Two-way ANOVA statistical methods to re-analyze our data (Fig3A/D, Fig4, Fig6A/B), and some of the results were different from those previously analyzed by Student's t test. We have re-marked the asterisk in the resubmitted figures based on the re-analyzed result and made corresponding modifications in the figure legends. Please

review the resubmitted manuscript and pictures.

Q8: In the last figure, why different numbers of ducks were used for viral DNA measurement in different tissues according to the graphs? The viral DNA was not tested in the indicated tissues in all infected ducks?

The author's response:

We are very sorry for our negligence in designing of experiment. Six infected ducklings per group were used for viral DNA and RNA measurement in Fig. 6A and Fig. 6C, and swab samples from 3 ducklings were collected for viral shedding determination. In the design of the experiment, we thought DNA loads in tissues and viral shedding were different indicators of pathogenicity, so we did not consider the use of the same number of duckling samples, which was our negligence. We can re-sample the cloaca for testing if necessary.

Review 2#:

The manuscript "Deletion double copies of US1 gene reduce infectivity of recombinant Duck Plague Virus in vitro and in vivo" by Wu et al. assessed the role of the US1 gene in the pathogenesis, replication, and gene expression of duck plague virus (DPV). The authors constructed two recombinant viruses, one with the deletion of the double US1 gene found in DPV, the other a revertant of this deletion. These recombinant viruses were compared in vitro and in vivo for their replication. The double deletion mutant resulted in reduced viral titers in growth kinetics, reduced number and size of plaques, and reduced copies of viral DNA in vitro. This mutant also resulted in reduced lesions and no mortality compared to the revertant and the wildtype control in ducks, with reduced titers in some organs. The manuscript provide interesting insight into the role of the US1 gene in DPV virulence, showing that its deletion results in an attenuated virus that has the potential to be used as a vaccine candidate. However, these findings are masked by the poor quality of the writing and often superficial description of details.

The author's response:

Thanks very much for your insightful comments and suggestions. They are valuable in improving the quality of our manuscript. According to your suggestions, we have carefully made a comprehensive revision and listed responses point-by-point as follows. Meanwhile, we resubmitted our documents with tracked changes that are highlighted in red.

Major comments :

Q1: The manuscript needs significant editing of the English language. Articles and prepositions are missing throughout the manuscript (e.g., the title should be

"Deletion of double copies of US1 gene reduces infectivity of recombinant Duck Plague Virus in vitro and in vivo"), and many sentences are confusing. A few examples are given in the specific comments below, but authors are encouraged to have the text revised by a native speaker or editing service.

The author's response:

Thanks for the kind comments and suggestions, they are very valuable in improving the quality of our manuscript. We attached great importance to every comment made by you. Therefore, we requested a professional service of English proofreading from American Journal Experts to revise the writing problems of this article. The detailed suggestion you mentioned in the title have been revised. The other grammar and statement errors have been carefully corrected in the revised manuscript. Meanwhile, we resubmitted our manuscript with tracked changes that are highlighted in red. We hope the revised manuscript can meet the language requirements of Microbiology Spectrum and you can consider giving this manuscript the opportunity to be published in this journal.

Q2: The manuscript is not very explicative to a broader audience, with many technical terms not described, and many abbreviations used without explanation (e.g., VZV, PRV, etc) or that need to be better defined, at least the most discussed ones (e.g., ICP22).

The author's response:

We gratefully thanks for the precious time the reviewer spent on our manuscript for making constructive remarks. According to your suggestions, we have carefully redefined all the abbreviations, please review the revised manuscript with tracked changes that are highlighted in red.

Q3: Although the methods seem to be sound, they are mostly superficially described, and many details are lacking and need to be expanded.

The author's response:

Thanks very much for your valuable suggestions, they are very valuable in improving the quality of our manuscript. Considering your suggestion, we have rewritten some parts of Materials and methods and added more details, please review the revised Materials and methods section in the resubmitted manuscript with tracked changes that are highlighted in red.

Q4: Additionally, pathological findings are described in a non-technical way, and should be revised by a pathologist to provide correct wording.

The author's response:

Thank you for pointing out this problem in manuscript. We sought help from a pathologist and rewrote the description of the pathological section under his guidance to ensure that it was correct and professional. Please review corresponding paragraph of the resubmitted manuscript with tracked changes that are highlighted in red.

Q5: In addition, some of the statistical analysis seem to be off, with (highly) significant statistical differences found for some graphs that don't seem to have such different values or have large variations (e.g., Figs 3A, 4, 6A).

The author's response:

Thank you so much for your careful check. In response to your question, We have retested differences between groups again using Student's t test. The test results were found to be the same as before. Considering our experimental data had at least three groups and other variables were involved, we used One-way and Two-way ANOVA statistical methods to re-analyze our data(Fig3A/D, Fig4, Fig6A/B), and some of the results were different from those previously analyzed by Student's t test. We have re-marked the asterisk in the resubmitted figures based on the re-analyzed result and made corresponding modifications in the figure legends. Please review the resubmitted manuscript and pictures.

Q6: The discussion is very confusing, and mostly superficial. Not much discussion about the genes that were tested..

The author's response:

Thanks very much for your insightful comments and suggestions. We have extensively revised the discussion section, including why we have to remove min-F cassette from DPV genome, the manipulation of DPV genome on genomic stability, and the genes that were tested, etc. Please review the revised manuscript with tracked changes that are highlighted in red(line 219-229, line 244-247 and line 299-311).

Specific comments

The abstract and the Importance are very contrasting - while the abstract emphasizes pathogenesis, the importance is solely focused on vaccine production. Although it is something that could result from this work, production (or testing) of a vaccine was not the point of this manuscript.

The author's response:

Thank you very much for your insightful comments and valuable suggestions. It really helped us to improve the quality of the manuscript. We agree with your comments and have rewritten the importance section of this manuscript, please review the red highlight content in the resubmitted manuscript.

L13-14: DPV possess a large genome consisting of 78 ORFs. Understanding the function and mechanism of each protein

L15-16: US1 is one of the two genes located in the repeat regions of DPV genome, but the function of its encoding protein in DPV

L17: Previous studies - please modify throughout

L21: that deletion of both copies

L22: could represent a potential candidate vaccine

L23: remove the word occurrence

L30: Change "can overcome the deficiency of them, which will be helpful for the epidemiological" to "can be differentiated from natural infection, which will be helpful for the epidemiological"

L32: available against DPV yet

L38: alternatively

L39: causative agent

L54: but is not an IE gene

L58: which can directly interact with cyclin-dependent kinase (CDK9) and inhibit the enzyme activity

L60: inhibits

L64: in involved in virus immune evasion

L79: The DPV genome includes a unique long (UL) and unique short (US) region - this description needs to be first mentioned in the previous paragraph

L82: US1 gene coexisting. The other duplicated gene is ICP4.

L84: we generated a deletion mutant with deletion of the double US1 genes - the way this mutant is termed is often confusing throughout the manuscript. Perhaps the best way to term it is "double US1 gene deletion mutant". I suggest using the abbreviation $2\Delta US1$ as often as possible.

L90: since results come first, the abbreviations should be spelled out first in the results

L94: an example of how the terms used for the mutant are not clear: "double copies of US1 gene deletion strain $2\Delta US1$ " What is the difference between the wild type and the revertant virus? It would be good to have an explanation of the rationale to have both.

L107: was used to indicate

L112: These results suggest

L124-125: wild type and revertant strains at every time point (Fig.3B). Plaque assays were performed to evaluate

L126: US1 gene resulted in smaller

L131: "indicating US1 gene promotes the transcription of DPV gene in all phases" - this is confusing. Which genes? Which phases?

L134: viral pathogenesis, and the flow chart of the animal experiment is shown in

L141: died

L145L post-inoculation

L146L collected and evaluated for gross and histopathological lesions.

L147: without any visible (or noticeable) lesions

L160: Livers of birds infected with

L161: Histopathological analysis of the spleen showed

L171: while DNA quantification of $2\Delta US1$ - measuring of genetic material does not necessarily correlate to replication, this was only measured at one time point, so there is no way to know if it increased.

L174: The deletion of US1 gene reduced the shedding rate

L188-189: An detailed explanation of the mini-F cassette should have been mentioned in the methods section, not only here.

L194: resulting in changes

L196: restore TK function to study the role

L200: viruses are purified

L206: by hiding

L212: tested

L217: but this hypothesis needs further verification

L224-225: had significantly reduced gross lesion

L226: completely avirulent

L236-238: This sentence does not relate much with what was found in the spleen, it should be placed before that sentence about the spleen.

L245-246: This sentence is not very coherent. Rephrase it.

The author's response:

Thank you very much for your insightful comments and valuable suggestions. We have carefully made a comprehensive revision based on your comment and suggestion, and the above writing problems you mentioned have been modified and checked by a professional service of English proofreading from American Journal Experts. Please review the red highlight content in the resubmitted manuscript.

L249: viral gene transcription in each tissue: This was true even for the spleen, despite no differences in titers. That is something that needs to be discussed.

The author's response:

Thank you very much for your insightful comments and valuable suggestions. Considering your suggestion, we have discussed that viral gene transcription reduced in spleen, please review the corresponding section with red highlighted track in the resubmitted manuscript(line 290-298).

L332: more details about animal experiments are needed - how were animals maintained, N, info about IACUC, species, etc.

The author's response:

Thank you very much for your insightful comments and valuable suggestions. We have added more details about animal species, numbers, maintained, ect. in the materials section. Please review the red highlight content in the resubmitted manuscript(line 420-425).

L337: was MEM administered to the Mock control?

The author's response:

Thank you very much for your insightful comments and valuable suggestions. Yes, MEM was used as a mock control for animal experiment, we have added a footnote of it. please review the resubmitted manuscript.

L338: When were swabs and tissues collected and what tissues?

The author's response:

Thank you very much for your insightful comments and valuable suggestions. As shown in Fig5A, 24 ducklings (six per group) were sacrificed at 5 days post infection, swabs and tissues(liver, spleen and duodenum) were collected at this time point for the lesions, viral loads and gene expression analysis. We have added the details in materials, please review the resubmitted manuscript in highlighted content(line 429-433).

L341: from cells or tissues were isolated.

L357-358: remove the repeated sentence: For extractions of RNA in organs, tissues frozen in liquid nitrogen should be ground into powder with grinding steel balls at first

The author's response:

Thank you very much for your insightful comments and valuable suggestions. We have carefully made a comprehensive revision based on your comment and suggestion, and the above writing problems you mentioned have been modified and checked by a professional service of English proofreading from American Journal Experts. Please review the red highlight content in the resubmitted manuscript.

June 28, 2022

Prof. Anchun Cheng
Key Laboratory of Animal Disease and Human Health of Sichuan Province, Sichuan Agricultural University
College of Veterinary Medicine
No. 211, Huimin Road
Wenjiang, Chengdu City, Sichuan, 611130, P.R. China, Sichuan 611130
China

Re: Spectrum01140-22R1 (Deletion double copies of US1 gene reduce infectivity of recombinant Duck Plague Virus in vitro and in vivo)

Dear Prof. Anchun Cheng:

Thank you for submitting your manuscript to Microbiology Spectrum. In general, both reviewers found the revised manuscript to be significantly improved. However each reviewer found additional points that need to be addressed. When submitting the revised version of your paper, please provide (1) point-by-point responses to the issues raised by the reviewers as file type "Response to Reviewers," not in your cover letter, and (2) a PDF file that indicates the changes from the original submission (by highlighting or underlining the changes) as file type "Marked Up Manuscript - For Review Only". Please use this link to submit your revised manuscript - we strongly recommend that you submit your paper within the next 60 days or reach out to me. Detailed instructions on submitting your revised paper are below.

Link Not Available

Sincerely,

Clinton Jones

Journals Department
Reviewer comments:

Reviewer #1 (Comments for the Author):

The manuscript has been significantly improved. The authors addressed all of my concerns except one. It was asked if US1 deletion affects the expression of neighboring genes US8 and/or US10. This was not answered. They sequenced US8 and US10 and found no alteration, which was expected. But I was asking about their gene (mRNA) expression.

Reviewer #2 (Comments for the Author):

The revised article is significantly improved, much clearer and better-written. The English language improved considerably. The authors have addressed my overall concerns. However, due to the extensive number of changes done to the manuscript, there were some minor concerns found as listed below.

L86-87: suggesting that it could be a promising candidate vaccine strain that can help control duck plague outbreaks.

L92-96: although the description is much improved, this sentence is too long and should be broken up in two or more sentences.

L173: showed obvious pathological findings. The term symptoms is not correct for animals, and here it is not description of clinical signs, but of pathological findings.

L179-180: After 5 days post inoculation, the 2 Δ US1 showed mild inflammatory...

L187: The duodenum showed mild necrosis in the mucosal epithelium

L198: Surprisingly, we found no significant difference in viral loads

L200-202: This sentence is confusing and a bit overinterpreting. What can be said by results is that 2 Δ US1 showed reduced copy numbers in cloacal samples, suggesting lower shedding of virus. By looking at copy numbers alone it cannot be inferred that infection (colonization) was reduced and clearance increased.

L213-216: This sentence about the constructions is very lengthy and should be broken up. This is true for many other sentences in the discussion (e.g., L230-233, L244-248, L248-252, L255-259,

L253-255: This sentence is confusing and not well-written.

L280: 2 Δ US1 showed attenuated virulence

L282: "but were not completely avirulent to the tested ducklings" - this sentence is not clear, do the authors mean "although 2 Δ US1 was not completely avirulent in inoculated ducklings"?

L285: what does "ocular lesions of the liver and thymus" mean? Is this referring to macroscopic or gross lesions?

L335: earlier by our group

Staff Comments:

Preparing Revision Guidelines

Please return the manuscript within 60 days; if you cannot complete the modification within this time period, please contact me. If you do not wish to modify the manuscript and prefer to submit it to another journal, please notify me of your decision immediately so that the manuscript may be formally withdrawn from consideration by Microbiology Spectrum.

Review 1#:

The manuscript has been significantly improved. The authors addressed all of my concerns except one. It was asked if US1 deletion affects the expression of neighboring genes US8 and/or US10. This was not answered. They sequenced US8 and US10 and found no alteration, which was expected. But I was asking about their gene (mRNA) expression.

The author's response:

We thank the reviewer for reading our manuscript carefully and giving the above positive comments for our revised article. According to your suggestion, we detected the expression of neighboring genes US10 and US8 by RT-qPCR at different time points after infection. As expected, the expression of US10 and US8 genes in the 2 Δ US1 strain decreased significantly at the early stage of infection compared with that of the wild strain, but gradually recovered as the infection time progressed. Combined with the results of sequencing, we believed that the decreased expression levels of US10 and US8 genes were caused by the transcriptional regulation of US1 gene, and had nothing to do with its genome stability. The results were attached in the supplementary figures (Figure S1). Meanwhile, we resubmitted our manuscript with tracked changes that are highlighted in red.

Review 2#:

The revised article is significantly improved, much clearer and better-written. The English language improved considerably. The authors have addressed my overall concerns. However, due to the extensive number of changes done to the manuscript, there were some minor concerns found as listed below.

The author's response:

Thanks very much for your positive comments for our revised article and your valuable suggestions. They are valuable in improving the quality of our manuscript. According to your suggestions, we have carefully made a comprehensive revision and listed responses point-by-point as follows. Meanwhile, we resubmitted our documents with tracked changes that are highlighted in red.

minor concerns :

L86-87: suggesting that it could be a promising candidate vaccine strain that can help control duck plague outbreaks.

L92-96: although the description is much improved, this sentence is too long and should be broken up in two or more sentences.

L173: showed obvious pathological findings. The term symptoms is not correct for animals, and here it is not description of clinical signs, but of pathological findings.

L179-180: After 5 days post inoculation, the 2 Δ US1 showed mild inflammatory...

L187: The duodenum showed mild necrosis in the mucosal epithelium

L198: Surprisingly, we found no significant difference in viral loads

L200-202: This sentence is confusing and a bit overinterpreting. What can be said by results is that 2ΔUS1 showed reduced copy numbers in cloacal samples, suggesting lower shedding of virus. By looking at copy numbers alone it cannot be inferred that infection (colonization) was reduced and clearance increased.

L213-216: This sentence about the constructions is very lengthy and should be broken up. This is true for many other sentences in the discussion (e.g., L230-233, L244-248, L248-252, L255-259,

L253-255: This sentence is confusing and not well-written.

L280: 2ΔUS1 showed attenuated virulence

L335: earlier by our group

The author's response:

Thank you very much for your insightful comments and valuable suggestions. We have carefully made a comprehensive revision based on your comment and suggestion, and the above writing problems you mentioned have been modified and checked. Please review the red highlight content in the resubmitted manuscript.

L282: "but were not completely avirulent to the tested ducklings" - this sentence is not clear, do the authors mean "although 2ΔUS1 was not completely avirulent in inoculated ducklings"?

The author's response:

Thank you very much for your insightful comments and valuable suggestions. We are sorry for the inappropriate statement. To make it clear, we have changed the sentence to "but it was still mildly pathogenic to the ducklings". Please review line 283 of the resubmitted manuscript.

L285: what does "ocular lesions of the liver and thymus" mean? Is this referring to macroscopic or gross lesions?

The author's response:

Thank you very much for your insightful comments and valuable suggestions. Yes, the sentence "ocular lesions of the liver and thymus" mean the macroscopic lesions. We have changed the sentence to "macroscopic lesions" to make it clear. Please review line 286 of the resubmitted manuscript.

September 5, 2022

Prof. Anchun Cheng
Key Laboratory of Animal Disease and Human Health of Sichuan Province, Sichuan Agricultural University
College of Veterinary Medicine
No. 211, Huimin Road
Wenjiang, Chengdu City, Sichuan, 611130, P.R. China, Sichuan 611130
China

Re: Spectrum01140-22R2 (Deletion of double copies of the US1 gene reduces the infectivity of recombinant duck plague virus in vitro and in vivo)

Dear Prof. Anchun Cheng:

Thank you for submitting your manuscript to Microbiology Spectrum. When submitting the revised version of your paper, please provide (1) point-by-point responses to the issues raised by the reviewers as file type "Response to Reviewers," not in your cover letter, and (2) a PDF file that indicates the changes from the original submission (by highlighting or underlining the changes) as file type "Marked Up Manuscript - For Review Only". It is crucial to address ALL concerns of Reviewer #1. Please use this link to submit your revised manuscript - we strongly recommend that you submit your paper within the next 60 days or reach out to me. Detailed instructions on submitting your revised paper are below.

Link Not Available

Sincerely,

Clinton Jones

Journals Department
Reviewer comments:

Reviewer #1 (Comments for the Author):

Genes in herpesvirus genomes are known to be tightly packed. That's why we have to be careful with the interpretation of gene deletion mutants, which can result in the abrogation of the expression of not only the deleted gene but also the neighboring genes due to the deletion of their promoters embedded in the body of the deleted gene. In the first and the second review it was asked if US1 deletion impacts US8 and US10 expression, which can affect the interpretation of the phenotype of CHv-2ΔUS1. As the authors show in the revision, US1 deletion does abrogate the expression of US10 and US8. Previous studies have shown that US8 and US10 deletions significantly reduce DPV virus production. Thus, the reduced replication of US1 deletion

virus may not be due to the lack of US1 but the reduced expression of US8 and US10. This changes the interpretation of their data and the conclusion of the manuscript.

Also, I disagree that the expression of US8 and US10 gradually recovers over time during infection. Fig S1 does not show it. What is CHv- Δ ICP22 in Fig S1? The figure should be about CHv-2 Δ US1.

Reviewer #2 (Comments for the Author):

The authors have addressed all my comments.

Staff Comments:

Preparing Revision Guidelines

Please return the manuscript within 60 days; if you cannot complete the modification within this time period, please contact me. If you do not wish to modify the manuscript and prefer to submit it to another journal, please notify me of your decision immediately so that the manuscript may be formally withdrawn from consideration by Microbiology Spectrum.

Reviewer #1 (Comments for the Author):

Genes in herpesvirus genomes are known to be tightly packed. That's why we have to be careful with the interpretation of gene deletion mutants, which can result in the abrogation of the expression of not only the deleted gene but also the neighboring genes due to the deletion of their promoters embedded in the body of the deleted gene. In the first and the second review it was asked if US1 deletion impacts US8 and US10 expression, which can affect the interpretation of the phenotype of CHv-2ΔUS1. As the authors show in the revision, US1 deletion does abrogate the expression of US10 and US8. Previous studies have shown that US8 and US10 deletions significantly reduce DPV virus production. Thus, the reduced replication of US1 deletion virus may not be due to the lack of US1 but the reduced expression of US8 and US10. This changes the interpretation of their data and the conclusion of the manuscript.

Also, I disagree that the expression of US8 and US10 gradually recovers over time during infection. Fig S1 does not show it. What is CHv-ΔICP22 in Fig S1? The figure should be about CHv-2ΔUS1.

The author's response:

Thanks very much for your positive comments for our revised article and your valuable suggestions. They are valuable in improving the quality of our manuscript. According to the literature, US1 has been reported as a general transcriptional regulatory protein in the homologues of herpesviruses, which regulates the expression of genes at various stages. Since US10 and US8 are adjacent genes of US1, we agree with your opinion that it is indeed possible that the deletion of US1 gene can result in the decreasing of neighboring genes due to the deletion of their promoters embedded in the body of the deleted gene, and the expression levels of other viral genes may also change accordingly. In view of this, it is difficult to accurately define the reason for the decrease in mRNA levels of US10 and US8 genes. Therefore, we designed the complement experiment of US10 and US8 to exclude the influence of US10 and US8 on the regulatory function of US1 gene. As a result, the overexpression of US10/US8/US10+US8 genes could not rescue the reduction of viral gene expression caused by US1 deletion except US10 and US8 (Miscellaneous File Not for Publication), which suggest that US10 and US8 gene has no relationship with the reduction of viral genes expression in 2ΔUS1. As you and literature mentioned, the decreased progeny virus production could be caused by US8 and US10 deletions. Therefore, we have modified the corresponding results and discussion section to make it more rigorous. In addition, the wrong figure legends you mentioned in Fig S1 have been modified, please review the revised manuscript with tracked changes that are highlighted in red.

Reviewer #2 (Comments for the Author):

The authors have addressed all my comments.

We thank the reviewer for reading our manuscript carefully and giving the above positive comments for our revised article.

The author's response:

We thank the reviewer for reading our manuscript carefully and giving the above positive comments for our revised article.

October 26, 2022

Prof. Anchun Cheng
Avian Disease Research Center, College of Veterinary Medicine, Sichuan Agricultural University
College of Veterinary Medicine
No. 211, Huimin Road
Chengdu City, Sichuan 611130
China

Re: Spectrum01140-22R3 (Deletion of double copies of the US1 gene reduces the infectivity of recombinant duck plague virus in vitro and in vivo)

Dear Prof. Anchun Cheng:

Your manuscript has been accepted, and I am forwarding it to the ASM Journals Department for publication. You will be notified when your proofs are ready to be viewed.

Sincerely,

Clinton Jones
Editor, Microbiology Spectrum
